# Rethinking Efficient Graph Coarsening via a *Non-Selfishness* Principle

Xu Bai [1]   Bin Lu [* 1]   Kun Zhang [2]   Shengbo Chen [3]   Xinbing Wang [1]   Chenghu Zhou [4]   Meng Jin [* 5]

## Abstract

Graph coarsening is a graph dimensionality reduction technique that aims to construct a smaller and more tractable graph while preserving the essential structural and semantic properties of the original graph. However, most existing methods rely on pair-wise similarity matching, where each node independently searches for its best partner based on global information. This *selfishness* matching paradigm incurs substantial computational and memory overhead. To address this problem, we shift to a *non-selfishness* principle that prioritizes the collective interference of neighborhood in coarsening, and propose an efficient method named NOPE, which achieves linear memory consumption and near-linear computational complexity in the number of nodes. Furthermore, we derive a faster variant NOPE*, which reduces $\mathcal{O}(\Delta \cdot d)$ interference evaluation to $\mathcal{O}(d)$ based on the local isotropy assumption, and consequently alleviates the computational bottleneck for high-degree nodes. Experimental results show that NOPE* achieves $1.8$–$10\times$ speedup over NOPE and surpass almost all baselines with 1-3 orders of magnitude acceleration. Meanwhile, learning on coarsened graphs yields comparable performance to original graphs, and can even show superior performance over LLM-based graph reasoning owing to compact graph information. The code can be available at https://github.com/dazonglian/NOPE-main.

## 1. Introduction

Graph-structured data is ubiquitous in real-world applications, and graph learning methods—such as graph neural networks (GNNs) (Corso et al., 2024) and large language models (LLMs) for graph reasoning (Jin et al., 2024; Zhao et al., 2026; Wu et al., 2026)—have achieved promising performance (Lu et al., 2024). However, the increasing scale and complexity of modern graphs impose substantial constraints for hardware and resources, limiting the practical applicability of existing approaches (Kumar et al., 2023b; Bai et al., 2024; Bai & Chen, 2025). Graph coarsening addresses this challenge through graph dimensionality reduction (Ni et al., 2026; Kumar et al., 2024). By aggregating nodes and edges into a smaller, more tractable graph, coarsening aims to preserve the essential structural and semantic properties of the original graph, enabling efficient analysis and downstream learning with minimal loss of information fidelity (Jin et al., 2022; Dickens et al., 2024; Kataria et al., 2025b).

Most existing graph coarsening methods adopt a pair-wise similarity matching paradigm, where nodes independently select merge partners based on similarity, e.g., FGC (Kumar et al., 2023a) merges nodes with high similarity by selecting pairs that minimize spectral loss, whereas A-CM (Dickens et al., 2024) conducts similarity-driven merging using representations obtained from a global graph convolution. These strategies are inherently *selfish*, as it considers only pair-wise affinity while neglecting the impact of merging on surrounding neighborhoods, which can distort neighborhood-level semantics, particularly in text-attributed graphs. Moreover, accurate similarity evaluation and partner search typically rely on global information, leading to substantial computational and memory overhead. These limitations motivate the need for a graph coarsening principle that moves beyond *selfish* pair-wise decisions while remaining both scalable and semantically faithful.

To this end, we propose a ***non-selfishness principle*** for graph coarsening, which prioritizes merges that minimize disturbance to local neighborhoods rather than relying solely on pair-wise similarity. To formalize this principle, we propose the neighborhood interference index ($\mathcal{I}$), a metric that quantifies the semantic deviation induced by a candidate merge. Guided by this index, we also develop **N**on-

---

[1]School of Information Science and Electronic Engineering, Shanghai Jiao Tong University, Shanghai, China [2]School of Environment Science and Engineering, Shanghai Jiao Tong University, Shanghai, China [3]School of Artificial Intelligence, Nanchang University, Nanchang, China [4]Institute of Geographic Sciences and Natural Resources Research, Chinese Academy of Sciences, Beijing, China [5]School of Artificial Intelligence, Shanghai Jiao Tong University, Shanghai, China. Correspondence to: Bin Lu <robinlu1209@sjtu.edu.cn>, Meng Jin <jinm@sjtu.edu.cn>.

*Proceedings of the $43^{rd}$ International Conference on Machine Learning*, Seoul, South Korea. PMLR 306, 2026. Copyright 2026 by the author(s).

selfishness **P**rinciple Graph Coars**E**ning (NOPE), a greedy interference-aware coarsening algorithm that achieves linear memory usage and near-linear time complexity. To further improve scalability as supernode degrees increase during coarsening, we derive a faster variant, NOPE*. It leverages a local isotropy assumption to reduce the per-merge evaluation cost from $\mathcal{O}(\Delta \cdot d)$ to $\mathcal{O}(d)$, where $\Delta$ denotes the dynamic average degree and $d$ denotes the feature dimension.

In summary, this work makes the following contributions.

1. Rethinking existing graph coarsening methods, we are the first to shift the paradigm to a *non-selfishness* principle, explicitly considering the minimization of neighborhood-level interference. Hereby, we propose NOPE, achieving a linear memory consumption and near-linear time complexity.

2. To further eliminating the increasing cost of degree-dependent interference quantification, we theoretically derive a faster variant NOPE*, reducing the per-merge complexity from $\mathcal{O}(\Delta \cdot d)$ to $\mathcal{O}(d)$.

3. Extensive experiments demonstrate that NOPE* achieves 1–3 orders of magnitude speedup over baselines while preserving or improving performance over both GNN- and LLM-based methods.

## 2. Related Work

**Structure Graph Coarsening**. The primary goals of structure graph coarsening algorithms are to reduce the computational burden of graph algorithms or to minimize graph storage (Chen et al., 2023b; Xu et al., 2024; Chu et al., 2024; Chen et al., 2023a). Their coarsening rules mainly focus on structural properties of graphs, such as preserving topological similarity between the coarsened graph and the original graph (Fan et al., 2012; Toivonen et al., 2011), or merging structurally similar nodes to reduce the number of edges and achieve compact graph representations (Yong et al., 2021; Liu et al., 2018; LeFevre & Terzi, 2010). However, these methods operate solely at the structural level and ignore node features or textual attributes, making them unsuitable for our setting.

**Feature Graph Coarsening**. For feature graph coarsening, FGC (Kumar et al., 2023a) first introduce $\epsilon - similarity$ to quantify feature consistency between the coarsened graph and the original graph, and formulate the problem as an optimization objective. Building on this idea, UGC (Kataria et al., 2024) improve efficiency by adopting a hash-based grouping strategy to merge similar nodes, and further propose AH-UGC (Kataria et al., 2025a), which enables continuous graph coarsening. Moreover, several GNN-based methods accelerate GNN computation through graph coars-

ening. MPG (Joly & Keriven, 2024), A-CM (Dickens et al., 2024), and CoCoA (Han et al., 2025) design coarsening criteria aligned with GNN-specific properties, such as message propagation similarity, to guide the coarsening process.

**Graph Condensation vs. Feature Graph Coarsening**. It should be noted that some graph condensation methods (Gao et al., 2025; Wang et al., 2024; Jin et al., 2022), although also aimed at enhancing the scalability of GNNs, do not strictly perform graph coarsening in a technical sense. In summary, the following issues distinguish graph condensation from graph coarsening: 1. **Lack of universality**. These methods optimize objectives tailored to specific GNN models, which limits their applicability to downstream models and often produces condensed graphs that do not preserve essential graph structures and features. 2. **Black-box compression**. The condensed graphs are typically learned without explicit node or edge correspondences to the original graphs, resulting in limited interpretability. In contrast, graph coarsening focuses on preserving the structural properties of the graph itself, yielding higher universality across downstream models and maintaining clear mappings between the original and coarsened graphs.

## 3. Background

### 3.1. Notations

Denote a text-attributed graph as $\mathcal{G} = (\mathcal{V}, \mathcal{E}, \mathcal{X}, \mathcal{R})$, where $\mathcal{V}$ denotes the node set, $\mathcal{E}$ is the edge set, $\mathcal{X} \in \mathbb{R}^{n \times d}(n = |\mathcal{V}|, d > 0)$ denotes the feature matrix and $\mathcal{R} = \{R_1, R_2, ..., R_{|\mathcal{V}|}\}$ represents the collection of raw text sequences associated with each node $v_i \in \mathcal{V}$. Each node $v_i$ is further assigned a ground-truth label $y_i \in \mathcal{Y}$. A complete summary of all notations used in this paper is provided in Appendix 3.

### 3.2. Problem Formulation

Given an original text-attributed graph $\mathcal{G} = (\mathcal{V}, \mathcal{E}, \mathcal{X}, \mathcal{R})$, the goal of text-attributed graph coarsening is to construct a compressed graph $\mathcal{G}^c = (\mathcal{V}^c, \mathcal{E}^c, \mathcal{X}^c, \mathcal{R}^c)$ with a minimal number of supernodes $\mathcal{V}^c$ ($|\mathcal{V}^c| \ll |\mathcal{V}|, |\mathcal{V}^c| = n^c$) and corresponding edges $\mathcal{E}^c$, while each supernode $\mathcal{V}_i^c$ is related to concise text $\mathcal{R}_i^c$. The relationship between the nodes $\mathcal{V}$ in graph $\mathcal{G}$ and the supernodes $\mathcal{V}^c$ in $\mathcal{G}^c$ can be represented by a mapping matrix $\mathbf{C}$, which belongs to the following set

$$\mathcal{C} = \left\{ \mathbf{C} \in \{0,1\}^{n^c \times n} \,\middle|\, \begin{array}{l} \langle \mathbf{C}_i, \mathbf{C}_j \rangle = 0, \forall i \neq j, \\ \|\mathbf{C}_k\|_0 \geq 1, \forall k \in \{1, \ldots, n^c\} \end{array} \right\}$$

where $\mathbf{C}_i$ and $\mathbf{C}_j$ represent $i$-th and $j$-th row of mapping matrix $\mathbf{C}$, $\langle \mathbf{C}_i, \mathbf{C}_j \rangle = 0$ means each original node must belong to and only belong to one supernode. $\|\mathbf{C}_k\|_0 \geq 1$ denotes each supernode contains at least one original node. Additionally, we set $r = (1 - \frac{n_c}{n})$ as the coarsening rate.

The node features $\mathcal{X} = \{\mathbf{x}_1, \mathbf{x}_2, ..., \mathbf{x}_n\}$ are encoded from the raw text $\mathcal{R}$ via Sentence-BERT. For each supernode $v_i^c$, its feature representation $\mathbf{x}_i^c$ is obtained by averaging the embeddings of all its constituent nodes: $\mathbf{x}_i^c = \frac{1}{|v_i^c|} \sum_{j \in v_i^c} \mathbf{x}_j$.

## 4. Methodology

This section presents a graph coarsening framework grounded in a *non-selfishness* principle. We define the neighborhood interference index ($\mathcal{I}$) to quantify the semantic disturbance induced by node aggregation. Based on it, we further propose a greedy coarsening algorithm, NOPE, which achieves near-linear time complexity and linear space complexity with respect to the number of nodes. To avoid exhaustive neighbor-wise similarity evaluations during coarsening, we introduce a faster variant, NOPE*, which leverages a local isotropy assumption to enable a constant-time approximation of neighborhood similarity.

### 4.1. Neighborhood Interference Index

Here, we first introduce the neighborhood interference index ($\mathcal{I}$). Considering a coarsening operation where nodes $v_p^c$ and $v_q^c$ are merged to a new supernode $v_w^c$, the interference caused to their combined neighborhood is quantified as

$$\mathcal{I}_{pq} = \sum_{i \in \mathcal{N}_p \cup \mathcal{N}_q} |v_p^c|(s_{ip} - s_{iw})^2 + |v_q^c|(s_{iq} - s_{iw})^2.$$

Here, the terms $(s_{ip} - s_{iw})^2$ and $(s_{iq} - s_{iw})^2$ quantify the information shift induced by merging. Specifically, they measure how much the original relationship between a neighbor $v_i^c$ and node $v_p^c$ (or $v_q^c$) deviates from its relationship with the resulting supernode $v_w^c$. These deviations are aggregated over the union of the neighbor sets and weighted by the corresponding node sizes $|v_p^c|$ and $|v_q^c|$, thereby reflecting the relative impact of each node involved in the merge. By explicitly penalizing neighborhood-level information shifts, this index discourages *selfish* merging decisions that preserve only pairwise similarity while distorting the surrounding semantic structure, and instead favors *non-selfish* merges that maintain neighborhood-level semantic stability. In this work, we adopt the dot product as the similarity measure, i.e., $s_{ij} = \langle \mathbf{x}_i, \mathbf{x}_j \rangle$. Under this definition, $\mathcal{I}_{pq}$ can be written as

$$\mathcal{I}_{pq} = \sum_{i \in \mathcal{N}_p \cup \mathcal{N}_q} \frac{|v_p^c||v_q^c|}{|v_p^c| + |v_q^c|}(s_{ip} - s_{iq})^2. \tag{1}$$

The derivation process can be found in Appendix B.1 and B.2.

To further illustrate the effectiveness of *non-selfishness* strategies, we compare the typical *selfish* strategy from two complementary perspectives: semantic preservation, measured by Dirichlet Energy, and structural balance, assessed

via Average Edge Betweenness Centrality (Avg. EBC). For specific experimental settings, please refer to the Appendix B.3. Figure 1 demonstrates that the *non-selfishness* strategy consistently maintains higher Dirichlet Energy, directly confirming its effectiveness in preserving feature discriminability and preventing over-smoothing during coarsening. In contrast, the *selfishness* baseline gradually eliminates feature differences. From the structural perspective, the smooth evolution of Avg. EBC under *non-selfishness* merging indicates a balanced distribution of connectivity responsibility, preventing shortest-path flow from becoming overly concentrated on a small number of edges. In contrast, the sharp increase of Avg. EBC under *selfish* merging reflects severe connectivity load concentration caused by improper disconnections. From a community perspective, it further intensifies flow pressure on inter-community edges, increasing the risk of community collapse.

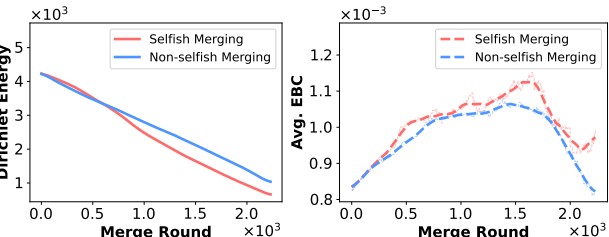

*Figure 1.* The changes of Dirichlet Energy and Average Edge Betweenness Centrality in the coarsened graphs based on *selfishness* and *non-selfishness* coarsening strategies.

### 4.2. *Non-selfishness* Principle Graph Coarsening

Motivated by our proposed neighborhood interference index $\mathcal{I}$, we design NOPE, an interference-aware greedy graph coarsening algorithm that iteratively contracts node pairs with minimal interference. To support efficient evaluation of $\mathcal{I}$ across iterations, NOPE precomputes the neighborhood interaction terms required for contraction decisions, and update and adjust them after each round of merging.

#### 4.2.1. NOPE ALGORITHM

As outlined in Algorithm 1, the execution flow is structured into four distinct phases: ❶ **Cache Initialization**. We initialize a cache $\Sigma$ to store, for each node, an aggregated measure of feature interactions with its neighborhood. This cache is designed to reuse intermediate results across iterations, thereby avoiding repeated computation of neighborhood statistics during the coarsening process. ❷ **Priority Queue Construction**. Based on the initialized cache, we compute $\mathcal{I}$ for each edge in the graph. All edge candidates are then inserted into a priority queue $\mathcal{H}$, which is ordered by increasing values of $\mathcal{I}$. This queue defines a greedy contraction order that prioritizes merges expected to introduce

minimal interference. ❸ **Main Coarsening Loop**. At each iteration, we extract the edge $(u, v)$ with the smallest interference index from $\mathcal{H}$ and merge the corresponding nodes into a new supernode $w$. The embedding of $w$ is computed as a size-weighted average of the embeddings of $u$ and $v$. To ensure efficiency, obsolete edge candidates generated by previous merges are handled using a lazy removal strategy. ❹ **Cache Updates**. After each merge operation, the cache $\Sigma$ is updated incrementally to reflect the modified graph structure. Here we adjust the cached statistics of the affected neighbors to account for the replacement of nodes $u$ and $v$ by the supernode $w$. This incremental update maintains exactness while reducing computational overhead.

### 4.2.2. NOPE TIME AND SPACE COMPLEXITY ANALYSIS

Assuming that the algorithm performs $\mathcal{O}(n)$ merge operations, NOPE has an overall time complexity of $\mathcal{O}(n \cdot \Delta^2 \cdot d)$, where $\Delta$ denotes the dynamic average node degree during the coarsening process. Since $\Delta \ll n$ in practical settings, the resulting time complexity is close to linear in the number of nodes. The space complexity is dominated by the storage of node features and the graph structure, requiring $\mathcal{O}(n_{\max}d + |\mathcal{E}|)$ memory, where $n_{\max} = 2n$. This leads to linear space complexity with respect to the input graph size and feature dimension, making the proposed method well suited for large-scale text-attributed graphs with high-dimensional features. A detailed complexity analysis is provided in Appendix C.

### 4.3. Faster *Non-selfishness* Principle Graph Coarsening

Although NOPE provides an effective coarsening strategy, the rapid growth of node degrees during merging incurs significant computational overhead due to the quadratic dependence of its time complexity on node degree. Under the assumption of local isomorphism, where adjacent node features exhibit minimal directional variation, we propose a faster variant, NOPE*. This variant replaces the exact importance measure $\mathcal{I}$ with its expectation $\mathbb{E}[\mathcal{I}]$, achieving substantially improved computational efficiency.

### 4.3.1. EXPECTED NEIGHBORHOOD INTERFERENCE

We first introduce the assumption of *local isotropy* (Athreya et al., 2018; Cai et al., 2021).

**Assumption 4.1** (*Local Isotropy*). Conditioned on the reference node $r$, the neighborhood embeddings satisfy

$$\mathbb{E}[\mathbf{x}_i] = 0, \quad \mathbb{E}[\mathbf{x}_i \mathbf{x}_i^T] = \sigma^2 \mathbf{I}_d, \quad \forall i \in \mathcal{N}_r,$$

where $\sigma^2 > 0$ denotes the variance associated with the reference node $r$ and $\mathbf{I}_d$ is the $d$-dimensional identity matrix.

Under Assumption 4.1, the expectation of $\mathcal{I}$ admits the

---

**Algorithm 1** NOPE

1: **Input:** Graph $\mathcal{G} = (\mathcal{V}, \mathcal{E}, \mathcal{X})$, Merge ratio $r$
2: **Output:** Coarsened graph structure and embeddings
3: Initialize $\mathbf{x} \in \mathbb{R}^{n_{max} \times d}$, $\mathbf{s} \in \mathbb{R}^{n_{max}}$, $\mathcal{A} \leftarrow \mathcal{V}$
4: $\mathbf{x}[0 : n] \leftarrow \mathcal{X}, \mathbf{s}[0 : n] \leftarrow \mathbf{1}_n$
5: Initialize sum-square cache $\Sigma \in \mathbb{R}^{n_{max}}$:
6: $\Sigma[i] \leftarrow \sum_{k \in \mathcal{N}_i} (\mathbf{x}_k^\top \mathbf{x}_i)^2 \quad \forall i \in \mathcal{V}$
7: Initialize Min-Heap $\mathcal{H}$
8: $\mathcal{H}.\text{push}((\mathcal{I}(u, v, \Sigma), u, v)) \quad \forall (u, v) \in \mathcal{E}$
9: **while** $|\mathcal{A}| > |\mathcal{V}| \cdot (1 - r)$ **do**
10: $\quad (\mathcal{I}_{uv}, u, v) \leftarrow \mathcal{H}.\text{pop}()$
11: $\quad$ **if** $u \notin \mathcal{A} \vee v \notin \mathcal{A}$ **then**
12: $\quad\quad$ **continue**
13: $\quad$ **end if**
14: $\quad$ Create new node $w$ with index $|\mathcal{V}| + 1$
15: $\quad$ Update node size: $\mathbf{s}_w \leftarrow \mathbf{s}_u + \mathbf{s}_v$
16: $\quad$ Update embedding: $\mathbf{x}_w \leftarrow (\mathbf{s}_u \mathbf{x}_u + \mathbf{s}_v \mathbf{x}_v)/\mathbf{s}_w$
17: $\quad$ Update neighbors: $\mathcal{N}_w \leftarrow (\mathcal{N}_u \cup \mathcal{N}_v) \setminus \{u, v\}$
18: $\quad$ **if** $\mathcal{N}_w \neq \emptyset$ **then**
19: $\quad\quad$ Let $\mathbf{x}_\mathcal{N} \in \mathbb{R}^{|\mathcal{N}_w| \times D}$ be neighbor feature matrix
20: $\quad\quad \mathbf{p}_w \leftarrow \langle \mathbf{x}_\mathcal{N}, \mathbf{x}_w \rangle, \ \mathbf{p}_u \leftarrow \langle \mathbf{x}_\mathcal{N}, \mathbf{x}_u \rangle, \ \mathbf{p}_v \leftarrow \langle \mathbf{x}_\mathcal{N}, \mathbf{x}_v \rangle$
21: $\quad\quad \Sigma[w] \leftarrow \|\mathbf{p}_w\|_2^2$
22: $\quad\quad$ Masks $\mathbf{m}_u \in \{0, 1\}^{|\mathcal{N}_w|}$ where $(\mathbf{m}_u)_k = \mathbf{1}_{k \in \mathcal{N}_u}$
23: $\quad\quad$ Masks $\mathbf{m}_v \in \{0, 1\}^{|\mathcal{N}_w|}$ where $(\mathbf{m}_v)_k = \mathbf{1}_{k \in \mathcal{N}_v}$
24: $\quad\quad \boldsymbol{\epsilon} \leftarrow \mathbf{p}_w^2 - (\mathbf{p}_u^2 \odot \mathbf{m}_u + \mathbf{p}_v^2 \odot \mathbf{m}_v)$
25: $\quad\quad \Sigma[\mathcal{N}_w] \leftarrow \Sigma[\mathcal{N}_w] + \boldsymbol{\epsilon}$
26: $\quad$ **end if**
27: $\quad$ Update $\mathcal{A} \leftarrow (\mathcal{A} \setminus \{u, v\}) \cup \{w\}$
28: $\quad$ **for** $k \in \mathcal{N}_w$ **do**
29: $\quad\quad \mathcal{H}.\text{push}(\text{Calculate } \mathcal{I}(w, k, \Sigma))$
30: $\quad\quad \mathcal{N}_k \leftarrow \mathcal{N}_k \setminus \{u, v\}, \mathcal{N}_k \leftarrow \mathcal{N}_k \cup \{w\}$
31: $\quad$ **end for**
32: **end while**

---

following closed-form expression

$$\mathbb{E}[\mathcal{I}] = \sum_{i \in \mathcal{N}_p \cup \mathcal{N}_q} \frac{|v_p^c||v_q^c|}{|v_p^c| + |v_q^c|} \sigma^2 \|\mathbf{x}_p - \mathbf{x}_q\|_2^2. \tag{2}$$

We denote $\mathbb{E}[\mathcal{I}]$ by $\mathcal{I}^*$ for brevity, the proof is provided in Appendix D.1. Compared to Eq. 1, Eq. 2 aggregates neighbor contributions through a shared-neighbor weighting term, retaining neighborhood information while avoiding neighbor-wise computations.

### 4.3.2. NOPE* ALGORITHM

The pseudo-code of NOPE* is provided in Algorithm 2 (Appendix D.2), Here we focus only on the key differences from Algorithm 1. Compared with Algorithm 1, Algorithm 2 replaces the exact measure $\mathcal{I}$ with its expectation $\mathcal{I}^*$, and updates the corresponding cache. This substitution col-

lapses explicit neighbor-wise dot-product computations into a single degree-weighted feature difference term, enabling efficient incremental updates during coarsening. As a result, the complexity of each merge evaluation is reduced from $O(\Delta \cdot d)$ to $O(d)$, leading to a substantial improvement in computational efficiency.

### 4.3.3. NOPE* TIME AND SPACE COMPLEXITY ANALYSIS

As discussed above, NOPE* reduces the cost of updating $\mathcal{I}$ for a single merge to $\mathcal{O}(d)$. Over $\mathcal{O}(n)$ merge operations, the overall time complexity is therefore approximately $\mathcal{O}(n \cdot \Delta \cdot d)$. Hence, the dependence on node degree is reduced from quadratic to linear, significantly improving scalability. This allows the generated dense graph to remain tractable during the coarsening process and enables the algorithm to be applied to relatively extreme coarsening scenarios.

In terms of memory consumption, NOPE* maintains a linear space complexity of $\mathcal{O}(n_{\max}d + |\mathcal{E}|)$, without introducing intermediate neighbor-level buffers. Compared to NOPE, it reduces the peak memory consumption. A detailed complexity analysis is provided in Appendix D.3.

### 4.3.4. MERGE PROCESS ANALYSIS

In this section, we investigate how NOPE and its faster variant NOPE* behave along the coarsening trajectory. Figure 2 reports the neighborhood interference index $\mathcal{I}$ across merge rounds on Citeseer.

In the early stage, the EWMA (Gardner Jr, 1985) trends of $\mathcal{I}$ under NOPE* closely match those of NOPE, indicating that $\mathcal{I}^*$ provides an accurate surrogate for $\mathcal{I}$ when local isotropy (Assumption 4.1) holds . As coarsening proceeds, the two curves gradually diverge and $\mathcal{I}$ under NOPE* increases more rapidly, suggesting growing approximation bias as merges become larger and more heterogeneous.

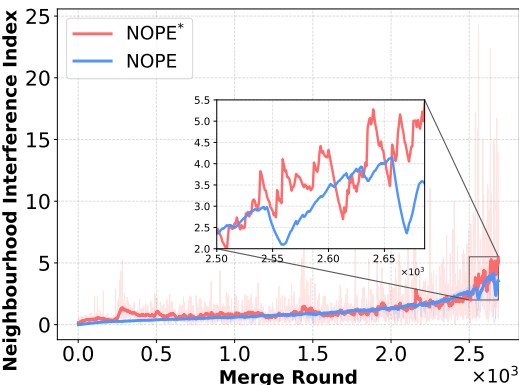

*Figure 2.* The neighborhood interference index $\mathcal{I}$ measured at each round of the NOPE and NOPE*.

Taken together, NOPE* is most effective during the early-to-

middle coarsening stages, where it achieves interference levels comparable to NOPE, or only mildly higher. As the merger progresses, NOPE* transitions into a higher-risk regime, characterized by increasing surrogate bias. In summary, our results suggest that applying NOPE at low to moderate coarsening rates is preferable.

## 5. Experiments

In this section, we first introduce the experimental settings, including the datasets, evaluation metrics, and baselines. Then, we evaluate our proposed NOPE and NOPE*, with extensive experiments to answer the following research questions (RQs):

- **RQ1**: How do NOPE and NOPE* compare to other baselines in terms of runtime and memory consumption?

- **RQ2**: In node classification tasks, how do the coarsened graphs generated by NOPE and NOPE* perform compared to those from other baseline methods?

- **RQ3**: How do the inference results on the coarsened graph generated by the NOPE and NOPE* models differ across different coarsening ratios?

### 5.1. Experimental Settings

**Datasets**. To evaluate the performance, we conduct node-level classification tasks on 6 real-world datasets, Citeseer (Hu et al., 2020), Ogb-Arxiv (Yang et al., 2016), Book (Wan & McAuley, 2018), and Products (Hu et al., 2020; He et al., 2024; Feng et al., 2024). The detailed information is summarized in Appendix E.

**Baseline**. We compare the proposed NOPE and NOPE* with four feature graph coarsening algorithms, including two graph algorithms that guarantee coarsening performance via $\epsilon - similarity$ (Kumar et al., 2023a) theory: FGC (Kumar et al., 2023a), UGC (Kataria et al., 2024), and two that seek to preserve downstream GNN performance: MPG (Joly & Keriven, 2024), A-CM (Dickens et al., 2024). We also benchmarked performance against classical node classification models based on GNNs, including GCN (Kipf & Welling, 2016), GIN (Xu et al., 2018) and GraphSAGE (Hamilton et al., 2017), while referencing conventional neighborhood sampling methods such as Random sampling (Wu et al., 2025; Chen et al., 2024), Degree sampling (Cao et al., 2024), and RAG sampling (Li et al., 2025). We aim for the algorithm's performance on coarsened graphs to approximate as closely as possible that of the original graph. For detailed model introduction and parameter settings, please refer to Appendix F.

**Evaluation Methods**. For the LLM-based node classification task, we select 10 nodes and employ LLMs for text

*Table 1.* Comparison of running memory and time on different datasets. Among them, OOM (out of memory) represents that the memory usage caused by the algorithm is greater than 96G, and OOT (out of time) indicates that the running time exceeds 5 hours.

| r= 0.5 | | Feature Graph Coarsening | | | | Ours | |
|---|---|---|---|---|---|---|---|
| | | FGC | MPG | UGC | A-CM | NOPE | NOPE* |
| Citeseer | Running Memory | 187.93MB | 471.11MB | 59.50MB | 96.72MB | 17.50MB | **16.05MB** |
| | Runtime | 2113.41s | 89.07s | 1.30s | 4.09s | 0.97s | **0.35s** |
| Products(subset) | Running Memory | / | / | 4331.60MB | 1228.96MB | **240.62MB** | 243.06MB |
| | Runtime | OOT | OOT | 3min1.76s | 1min9.38s | 11.57s | **6.46s** |
| Ogb-Arxiv | Running Memory | OOM | OOM | 53,885.60MB | 6,526.96MB | **1,307.60MB** | 1,354.61MB |
| | Runtime | / | / | 37min17.51s | 16min1.00s | 5min12.79s | **1min4.38s** |
| Book | Running Memory | OOM | OOM | OOM | 27,351.26MB | 5,542.74MB | **4,837.80MB** |
| | Runtime | / | / | / | 1h21min6.50s | 44min48.80s | **4min29.46s** |
| Products | Running Memory | OOM | OOM | OOM | / | / | **35,348.38MB** |
| | Runtime | / | / | / | OOT | OOT | **1h50min9.36s** |

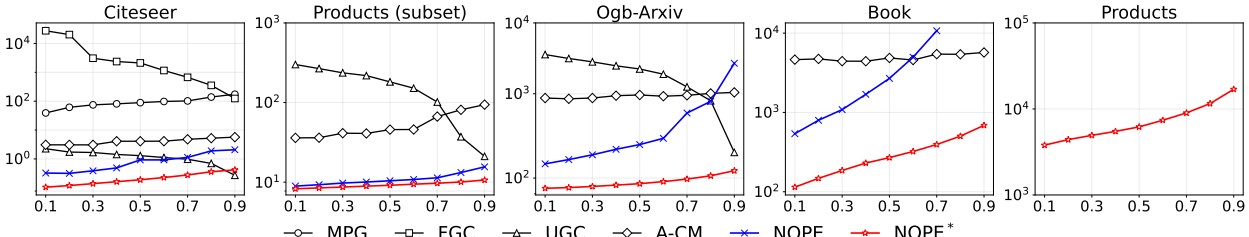

*Figure 3.* Time consumption in five datasets under different ratios $r$.

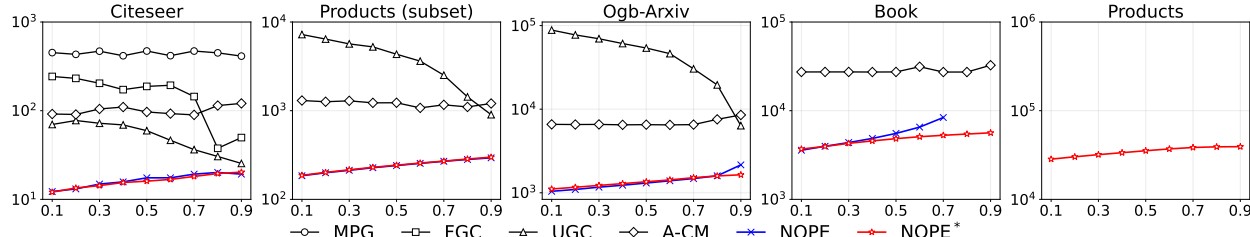

*Figure 4.* Memory consumption in five datasets under different ratios $r$.

reasoning to obtain corresponding text summaries for each supernode, i.e.

$$R_{v^c} = f_{\text{LLM}}\big(\text{Prompt}_{supernode}(\{R_i \mid v_i \in \mathcal{S}_{v^c}\},\ \theta)\big)$$

where $\mathcal{S}_{v^c} \subseteq v^c$, $|\mathcal{S}_{v^c}| = \min(|v^c|, 10)$ and $\theta$ denotes the parameters of the LLM. For single node label prediction, we jointly performed label prediction on both the target node's text and the text of its parent supernode, that is

$$\hat{y}_i = f_{LLM}(\text{Prompt}_{predict}(\{v_i, v^c|v_i \in v^c\}), \theta)$$

All inference tasks are conducted by Llama 3.1 8B-Instruct (Dubey et al., 2024).

For GNN-based node classification tasks, GNN message passing is first performed on the coarsened graph. Subsequently, predictions are made for target nodes using both their original features and the GNN-output features, i.e.:

$$\hat{y} = \text{MLP}_{classifier}(\text{MLP}_{node}(\mathbf{x}_i)||\text{GNN}_{(\mathcal{G}^c, \mathcal{X}^c)}(v_i)).$$

Here $\text{MLP}_{node}$ denotes the node feature encoder, $\text{GNN}_{(\mathcal{G}^c, \mathcal{X}^c)}$ refers to the output of the GNN model on the coarsened graph $\mathcal{G}^c$, and $\text{MLP}_{classifier}$ indicates the node label predictor. For specific prompt design and the implementation details, please refer to the Appendix G.

### 5.2. Runtime and Memory Consumption Analysis (RQ1)

Table 1 compares runtime and memory consumption under a fixed coarsening ratio of 0.5. Overall, NOPE and NOPE* consistently outperform existing graph coarsening methods.

On the small Citeseer, all methods are feasible, yet other approaches incur substantial computational overhead. In contrast, NOPE* achieves a 0.35s runtime with only 16MB

*Table 2.* LLM node classification results on different datasets under $r = 0.5$.

| Node Classification | | Full Graph | | | Graph Coarsening | | | | Ours | |
|---|---|---|---|---|---|---|---|---|---|---|
| **Dataset** | **Metric** | *Random* | *Degree* | *RAG* | FGC | MPG | UGC | A-CM | NOPE | NOPE* |
| Citeseer | ACC | 0.5924 | 0.5768 | 0.5956 | 0.5911 | 0.6112 | 0.5893 | 0.6081 | **0.6206** | 0.6018 |
| | F1 | 0.6098 | 0.5972 | 0.6147 | 0.6038 | 0.6280 | 0.6049 | 0.6221 | **0.6403** | 0.6138 |
| Products(subset) | ACC | 0.5761 | 0.5859 | 0.5789 | / | / | 0.5484 | 0.6755 | 0.6783 | **0.6790** |
| | F1 | 0.5980 | 0.6078 | 0.6003 | / | / | 0.5676 | 0.6754 | 0.6820 | **0.6827** |
| Ogb-Arxiv | ACC | 0.4029 | 0.4142 | 0.4256 | / | / | 0.3482 | **0.3925** | 0.3808 | 0.3795 |
| | F1 | 0.3940 | 0.4046 | 0.4154 | / | / | 0.3456 | **0.3818** | 0.3738 | 0.3737 |
| Book | ACC | 0.8996 | 0.8994 | 0.9080 | / | / | / | 0.9162 | 0.9170 | **0.9244** |
| | F1 | / | / | / | / | / | / | / | / | / |

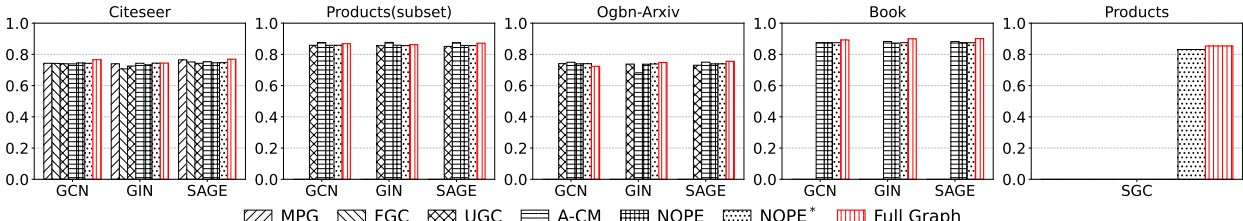

*Figure 5.* Accuracy/Hamming loss for GNN node classification in five datasets under $r = 0.5$.

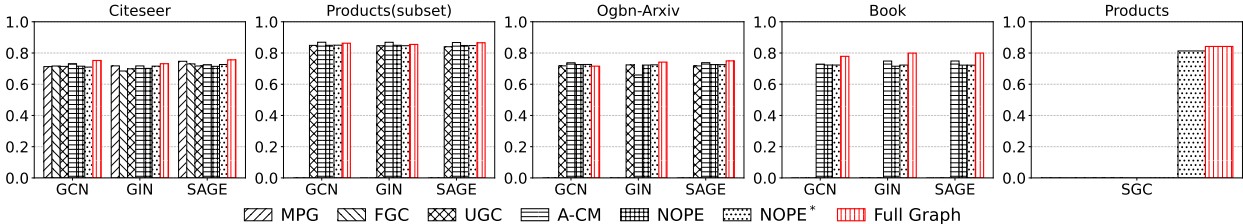

*Figure 6.* F1-score for GNN node classification in five datasets under $r = 0.5$.

memory, yielding orders-of-magnitude speedups over FGC and MPG. As graph scale increases, the performance gap becomes more pronounced. On Products(subset) and Ogb-Arxiv, several baselines fail due to OOT or OOM, whereas NOPE and NOPE* complete within seconds to minutes using significantly less memory (e.g., 1.35GB vs. 53GB on Ogb-Arxiv). On large-scale datasets (Book and Products), most baselines become infeasible; notably, NOPE* is the only method that successfully processes the full Products graph under the 96GB memory constraint.

Figures 3 and 4 further evaluate the time and memory efficiency under different coarsening ratios $r$. NOPE exhibits good time efficiency on most datasets at low and moderate coarsening ratios. However, as the coarsening rate increases, leading to degraded performance. Similar to the previous analysis, this effect arises from the significant increase in node degrees in the coarsened graph. In contrast, the faster variant NOPE* remains efficient and stable in all settings. In terms of memory consumption, NOPE and NOPE* consis-

tently achieve the lowest memory usage across all datasets and coarsening ratios, demonstrating strong memory efficiency and stability.

### 5.3. Node Classification Performance (RQ2)

#### 5.3.1. LLM-BASED NODE CLASSIFICATION

Table 2 summarizes the performance of different methods on LLM-based node classification under a coarsening ratio of 0.5. Overall, the performance of NOPE is significantly better than heuristic benchmark methods such as Random, Degree, and RAG. Even on the Products(subset), it achieves an increase of over 15%. This highlights the clear advantages of node merging with interference awareness in terms of semantic richness and fidelity. Compared with FGC and UGC, NOPE achieves improved results on all graphs, while NOPE* largely follows the same trend, suggesting that semantic information required by LLM-based classifiers can be effectively preserved even under lightweight

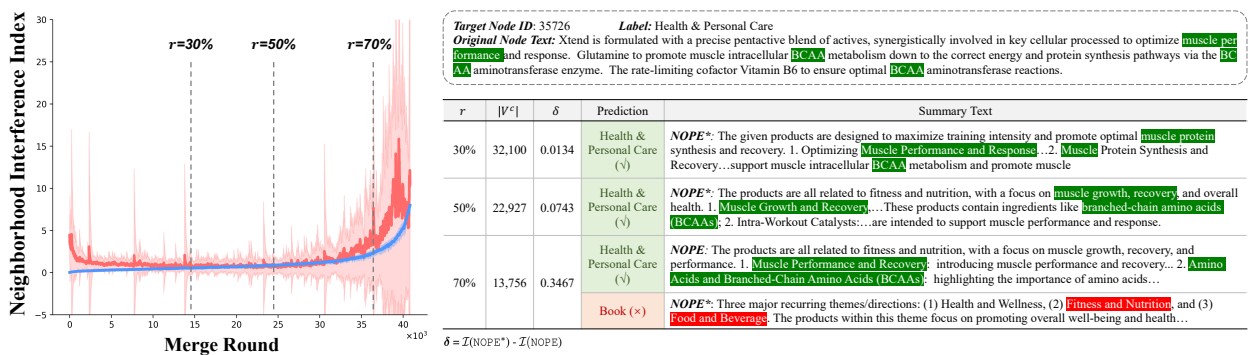

*Figure 7.* The case study on the Products dataset respectively presents the text of the supernode generated by the NOPE and NOPE* algorithms under different coarsening rates $r$, as well as the corresponding final prediction.

approximations. In addition, compared to GNN-specific graph coarsening methods such as MPG and A-CM, NOPE and NOPE* exhibit more stable performance across different graphs. Notably, A-CM achieves marginal improvements only on Ogb-Arxiv, where heuristic neighborhood retrieval is particularly effective, partially accounting for the competitiveness of similarity-based coarsening on this graph.

Results under coarsening ratios of 0.3 and 0.7, reported in Tables 7 and 8, exhibit consistent performance trends across methods. Notably, NOPE and NOPE* maintain competitive and stable performance under both settings, further demonstrating their robustness to varying coarsening rates.

### 5.3.2. GNN-BASED NODE CLASSIFICATION

Results for GNN-based node classification under a coarsening ratio of 0.5 are reported in Figures 5 and 6. Compared with full-graph GNN models, NOPE achieves performance that remains close to that obtained on the original graphs across most graphs, suggesting that the coarsened graphs retain a large portion of the information required for GNN inference. Relative to feature-based graph coarsening methods such as FGC and UGC, NOPE shows comparable performance in most settings, indicating that interference-aware coarsening is a viable alternative under GNN-based evaluation. Moreover, although MPG and A-CM are strong GNN-specific baselines tailored to exploit structural properties, NOPE, while being model-agnostic, achieves broadly comparable results and in some cases performs favorably on certain graphs. The efficient variant NOPE* generally follows similar trends, with modest variations in performance.

Additional results under coarsening ratios of 0.3 and 0.7 are presented in Figures 8, 9 and Figures 10, 11. Across different coarsening rates, the relative performance trends remain largely consistent, with NOPE and NOPE* exhibiting comparable behavior and alternating advantages across datasets, indicating stable performance under varying levels of graph reduction.

### 5.4. Case Study (RQ3)

In this section, we present a representative case to illustrate supernode text generation across different coarsening rates. Take a node (ID 35726, label: Health & Personal Care) from the Products(subset) as a case, the supernode summaries generated by NOPE and NOPE*, assessing their support for accurate label prediction.

Under low and moderate coarsening rates ($r = 0.3$ and $r = 0.5$), both NOPE and NOPE* produce semantically coherent summaries that align well with the target node, preserving key concepts related to muscle performance, recovery, and BCAAs. In these cases, the target category is correctly identified by both methods. At a high coarsening rate ($r = 0.7$), however, the two methods diverge in this example. While NOPE continues to capture the core semantics of the target node, NOPE* yields less focused and less discriminative summaries, resulting in an incorrect prediction. This behavior is accompanied by increased neighborhood interference, suggesting that approximation errors under aggressive coarsening may introduce semantically heterogeneous neighbors.

Overall, this case study highlights a representative failure mode of NOPE* at high coarsening rates. Nevertheless, quantitative results show that such cases are uncommon, and NOPE* remains accurate in the majority of scenarios.

## 6. Conclusion

In this work, we introduce a *non-selfishness* principle for graph coarsening that explicitly accounts for neighborhood-level interference during node aggregation. By formulating the neighborhood interference index ($\mathcal{I}$), we move beyond pairwise global similarity and enable neighborhood-aware merging that preserves local semantic consistency. Building on this principle, we propose NOPE, a greedy coarsening algorithm with near-linear time complexity and linear memory usage, along with its fast approximation NOPE*, which reduces the overhead of interference computation at

higher coarsening ratios. Extensive experiments on large text-attributed graphs show that `NOPE` and `NOPE`* achieve favorable efficiency–performance trade-offs for both LLM-based and GNN-based node classification, while significantly reducing runtime and memory consumption compared to existing methods.

Looking ahead, the interference-aware formulation provides a flexible foundation for future extensions, including dynamic or temporal graphs and scaling to billion-scale networks through streaming, distributed and so on.

## Acknowledgments

We would like to thank the reviewers for their suggestions to improve this paper. This work is supported by National Natural Science Foundation of China (No. T2421002, 62602003, 62272293) and Postdoctoral Fellowship Program of CPSF under Grant Number No. GZB20250806.

## Impact Statement

This paper presents work whose goal is to advance the field of Machine Learning. There are many potential societal consequences of our work, none which we feel must be specifically highlighted here.

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

*Table 3.* Notations and corresponding meanings.

| Notation | Meaning | Notation | Meaning |
|---|---|---|---|
| $\mathcal{G}$ | Text-attributed graph (TAG) | $\mathcal{V}$ | All nodes in TAG |
| $\mathcal{E}$ | All edges in TAG | $\mathcal{R}$ | The raw text sequences of nodes |
| $\mathcal{X}$ | The feature matrix of nodes | $n$ | The number of nodes in $\mathcal{G}$ |
| $d$ | The dimension of feature | $v_i$ | Node of $\mathcal{V}$ |
| $\mathcal{N}_{v_i}$ | The neighbor set of node $v_i$ | $\mathbf{x}_i/\mathcal{X}[v_i]$ | Feature vector of node $v_i$ |
| $\mathcal{G}_c$ | Coarsened text-attributed graph | $\mathcal{V}_c$ | All nodes in coarsened TAG |
| $v_i^c$ | Node in $\mathcal{G}_c$ | $u/v/w$ | Node in $\mathcal{G}$ or $\mathcal{G}_c$ |
| $\mathcal{E}_c$ | All edges in coarsened TAG | $|v_i^c|$ | Size of node set $v_i^c$ |
| $s_{ij}$ | Similarity between nodes $v_i$ and $v_j$ | $n_{max}$ | The number of pre-allocated nodes |
| $\mathbf{C}$ | The mapping matrix | $r$ | The coarsening rate |
| degree | The average degree of $\mathcal{G}$ | $\Delta$ | The dynamic average degree during coarsening |

## A. Notations and Corresponding Meanings

In this section, we supplement the paper with a detailed list of notations and their corresponding meanings, as listed in Table 3.

## B. Some Inference about $\mathcal{I}$.

For convenience, we consider the case where supernodes $v_p^c$ and $v_q^c$ are merged to form node $v_w^c$.

### B.1. Embedding Update

As seen in part 3.2 above, the embedding of node $v_w^c$ in the coarsened graph $\mathcal{G}_c$ is obtained by averaging the embeddings of its corresponding nodes in the original graph $\mathcal{G}$, i.e., $\mathbf{x}_w^c = \frac{1}{|v_w^c|} \sum_{j \in v_w^c} \mathbf{x}_j$. Hence, the embedding updating process can be formulated as

$$\mathbf{x}_w^c = \frac{1}{|v_w^c|} \sum_{i \in v_w^c} \mathbf{x}_w^c = \frac{1}{|v_p| + |v_q|} \left( \sum_{i \in v_p^c} \mathbf{x}_i + \sum_{j \in v_q^c} \mathbf{x}_j \right) = \frac{1}{|v_p^c| + |v_q^c|} \left( |v_p^c| \cdot \frac{1}{|v_p^c|} \sum_{i \in v_p^c} \mathbf{x}_i + |v_q^c| \cdot \frac{1}{|v_q^c|} \sum_{j \in v_q^c} \mathbf{x}_j \right)$$

$$= \frac{|v_p^c| \mathbf{x}_p^c + |v_q^c| \mathbf{x}_q^c}{|v_p^c| + |v_q^c|}.$$

### B.2. Simplification Process of Index $\mathcal{I}$

Given

$$\mathcal{I}_{pq} = \sum_{i \in \mathcal{N}_p \cup \mathcal{N}_q} |v_p|(s_{ip} - s_{iw})^2 + |v_q|(s_{iq} - s_{iw})^2,$$

where

$$s_{iw} = \langle \mathbf{x}_i, \mathbf{x}_r \rangle = \mathbf{x}_i^T \frac{|v_p^c| \mathbf{x}_p^c + |v_q^c| \mathbf{x}_q^c}{|v_p^c| + |v_q^c|} = \frac{|v_p^c| s_{ip} + |v_q^c| s_{iq}}{|v_p^c| + |v_q^c|};$$

$$s_{ip} - s_{iw} = \frac{|v_p^c| s_{ip} - |v_q^c| s_{iq}}{|v_p^c| + |v_q^c|}, \quad s_{iq} - s_{iw} = \frac{|v_p^c| s_{iq} - |v_q^c| s_{ip}}{|v_p^c| + |v_q^c|}.$$

Then

$$\mathcal{I}_{pq} = \sum_{i \in \mathcal{N}_p \cup \mathcal{N}_q} \left( \frac{|v_p^c||v_q^c|^2}{(|v_p^c| + |v_q^c|)^2} + \frac{|v_p^c|^2|v_q^c|}{(|v_p^c| + |v_q^c|)^2} \right)(s_{ip} - s_{iq})^2 = \sum_{i \in \mathcal{N}_p \cup \mathcal{N}_q} \frac{|v_p^c||v_q^c|}{|v_p^c| + |v_q^c|}(s_{ip} - s_{iq})^2.$$

### B.3. Experimental Design

To examine the impact of the *non-selfishness* principle, we conduct controlled experiments comparing a *non-selfishness* coarsening process with a *selfishness* counterpart that selects merge pairs solely based on cosine similarity. Both of these

methods employ the same greedy coarsening process, ensuring that any behavioral differences are solely due to the impact of whether a *non-selfishness* strategy is adopted. The experiment is conducted on the Citeseer citation network. During each merging round, the Dirichlet Energy of the coarsened graph and the Average Edge Betweenness Centrality (Avg. EBC) are recorded.

(1) Dirichlet Energy: a semantic measure of feature smoothness on the graph, where higher values indicate stronger feature variation and better preservation of high-frequency semantic information. Its specific form is

$$\text{Dirichlet Energy} = \frac{1}{|\mathcal{E}|} \sum_{(i,j)\in\mathcal{E}} ||\mathbf{x}_i - \mathbf{x}_j||_2^2.$$

(2) Average Edge Betweenness Centrality (Avg. EBC) is a structural measure that reflects how shortest-path connectivity is distributed across edges during coarsening, indicating whether connectivity responsibility remains evenly shared or becomes excessively concentrated on a small subset of critical edges, which may also manifest as premature degradation of community structure. It can be written as

$$\text{Avg EBC} = \frac{1}{|\mathcal{E}|} \sum_{e\in\mathcal{E}} \sum_{s,t\in\mathcal{V}, s<t} \frac{\sigma_{st}(e)}{\sigma_{st}},$$

where $\sigma_{st}$ represent the total number of shortest paths between nodes $s$ and $t$, and $\sigma_{st}(e)$ denotes the number of those shortest paths passing through edge $e$.

## C. Time and Space Complexity Analysis of `NOPE`

**Time Complexity**: Initializing the priority queue requires computing the index $\mathcal{I}$ for all edges. First, the cache items are pre-computed in $\mathcal{O}(n \cdot \text{degree} \cdot d)$, resulting in a total heap construction time of $\mathcal{O}(\mathcal{E}(\text{degree} \cdot d + \log \mathcal{E}))$. Under the assumption that $\mathcal{E} \approx \frac{1}{2} \cdot \mathcal{N} \cdot \text{degree}$, this complexity approximates to $\mathcal{O}(n \cdot \text{degree}^2 \cdot d)$. In the iterative phase, merging nodes generates approximately $\Delta$ new edges. For each new edge, computing $\mathcal{I}$ involves identifying common neighbors and calculating dot product differences, which incurs a cost of $\mathcal{O}(\Delta \cdot d)$. Consequently, the dominant cost per iteration arises from handling these new edges: $\Delta \cdot \mathcal{O}(\Delta \cdot d) = \mathcal{O}(\Delta^2 \cdot d)$.

Considering the algorithm performs $\mathcal{O}(n)$ merge operations, the overall complexity is approximate to $\mathcal{O}(n \cdot \Delta^2 \cdot d)$. In sparse graphs where $\Delta \ll n$, this approaches linearity with respect to graph size.

**Space Complexity**: The spatial requirements are primarily dictated by the storage of high-dimensional node features and the graph topology. In terms of feature storage, the algorithm utilizes a pre-allocation strategy to avoid memory fragmentation. The feature matrix vectors requires $\mathcal{O}(n_{max}d)$ space. Regarding graph topology, the adjacency list and the priority queue (min-heap) store graph connectivity, consuming $\mathcal{O}(|\mathcal{E}|)$ space. The auxiliary arrays for node sizes and the structural cache require $\mathcal{O}(n)$ space. Moreover, during the vectorized operations, temporary matrices of size $\mathcal{O}(\Delta \cdot d)$ are created, but these are transient and do not impact the asymptotic bound.

The overall space complexity is $\mathcal{O}(n_{max}d + |\mathcal{E}|)$. This is linear with respect to the input graph size, making the algorithm memory-efficient for large-scale, text-attributed graphs where $n$ and $d$ are large.

## D. Some Inference about `NOPE`* Algorithm

### D.1. Proof of the Expectation of $\mathcal{I}$ ($\mathbb{E}(\mathcal{I})$)

Let $\delta \triangleq \mathbf{x}_p - \mathbf{x}_q$, $\mathcal{N}_r = \mathcal{N}_p \cup \mathcal{N}_q$ and $M \triangleq \sum_{i\in\mathcal{N}_p\cup\mathcal{N}_q} \mathbf{x}_i\mathbf{x}_i^T \in \mathbb{R}^{d\times d}$, then the $\mathcal{I}_{pq}$ can be written as

$$\mathcal{I}_{pq} = \sum_{i\in\mathcal{N}_p\cup\mathcal{N}_q} \frac{|v_p^c||v_q^c|}{|v_p^c| + |v_q^c|}(s_{ip} - s_{iq})^2 = \sum_{i\in\mathcal{N}_p\cup\mathcal{N}_q} \frac{|v_p^c||v_q^c|}{|v_p^c| + |v_q^c|}((\mathbf{x}_i^c)^T(\mathbf{x}_p^c - \mathbf{x}_q^c))^2 = \frac{|v_p^c||v_q^c|}{|v_p^c| + |v_q^c|}\delta^T M\delta.$$

Given that the assumption 4.1, the expectation of $\mathbb{E}[\mathcal{I}]$ is

$$\mathbb{E}[\mathcal{I}] = \frac{|v_p^c||v_q^c|}{|v_p^c| + |v_q^c|}\delta^T\mathbb{E}[M]\delta = \frac{|v_p^c||v_q^c|\sigma^2}{|v_p^c| + |v_q^c|}|\mathcal{N}_r|||\mathbf{x}_p - \mathbf{x}_q||_2^2.$$

---

**Algorithm 2** NOPE*

---

1: **Input:** Graph $\mathcal{G} = (\mathcal{V}, \mathcal{E}, \mathcal{X})$, Merge ratio $r$
2: **Output:** Coarsened graph structure and embeddings
3: Initialize $\mathbf{x} \in \mathbb{R}^{n_{max} \times d}$, $\mathbf{s} \in \mathbb{R}^{n_{max}}$, $\mathbf{d} \in \mathbb{R}^{n_{max}}$, $\mathcal{A} \leftarrow \mathcal{V}$
4: $\mathbf{x}[0:n] \leftarrow \mathcal{X}$, $\mathbf{s}[0:n] \leftarrow \mathbf{1}_n$, $\mathbf{d}[i] \leftarrow |\mathcal{N}_i| \quad \forall i \in \mathcal{V}$
5: Initialize Min-Heap $\mathcal{H}$
6: $\mathcal{H}$.push$((\mathcal{I}^*(u,v), u, v)) \quad \forall (u,v) \in \mathcal{E}$
7: **while** $|\mathcal{A}| > |\mathcal{V}| \cdot (1-r)$ **do**
8: $\quad (\mathcal{I}^*(u,v), u, v) \leftarrow \mathcal{H}$.pop()
9: $\quad$ **if** $u \notin \mathcal{A} \vee v \notin \mathcal{A}$ **then**
10: $\quad\quad$ **continue**
11: $\quad$ **end if**
12: $\quad$ Create new node $w$ with index $|\mathcal{V}| + 1$
13: $\quad$ Update node size: $\mathbf{s}_w \leftarrow \mathbf{s}_u + \mathbf{s}_v$
14: $\quad$ Update embedding: $\mathbf{x}_w \leftarrow (\mathbf{s}_u \mathbf{x}_u + \mathbf{s}_v \mathbf{x}_v)/\mathbf{s}_w$
15: $\quad$ Update neighbors: $\mathcal{N}_w \leftarrow (\mathcal{N}_u \cup \mathcal{N}_v) \setminus \{u, v\}$
16: $\quad$ Update degree: $\mathbf{d}[w] \leftarrow |\mathcal{N}_w|$
17: $\quad$ Update $\mathcal{A} \leftarrow (\mathcal{A} \setminus \{u, v\}) \cup \{w\}$
18: $\quad$ **for** $k \in \mathcal{N}_w$ **do**
19: $\quad\quad \mathcal{N}_k \leftarrow (\mathcal{N}_k \setminus \{u, v\}) \cup \{w\}$
20: $\quad\quad$ Update neighbor degree $\mathbf{d}[k] \leftarrow |\mathcal{N}_k|$
21: $\quad\quad w_{edge} \leftarrow (\mathbf{s}_w \cdot \mathbf{s}_k)/(\mathbf{s}_w + \mathbf{s}_k)$
22: $\quad\quad d_{struct} \leftarrow \mathbf{d}[w] + \mathbf{d}[k] - |\mathcal{N}_w \cap \mathcal{N}_k| - 2$
23: $\quad\quad d_{feat} \leftarrow \|\mathbf{x}_w - \mathbf{x}_k\|_2^2$
24: $\quad\quad \mathcal{H}$.push$((w_{edge} \cdot w_{struct} \cdot w_{feat}), w, k))$
25: $\quad$ **end for**
26: **end while**

---

### D.2. The Pseudocode of NOPE*

The pseudocode of NOPE* can be found in Algorithm 2.

### D.3. Time and Space Complexity Analysis of NOPE*

**Time Complexity**: The initialization phase involves computing $\mathcal{I}^*$ for all existing edges. and it is computed as the Euclidean distance with a cost of $\mathcal{O}(d)$. Consequently, constructing the initial min-heap takes $\mathcal{O}(\mathcal{E}(d + log\mathcal{E}))$, which is dominated by the feature distance calculation, resulting in $\mathcal{O}(\mathcal{E}d) \approx \mathcal{O}(n \cdot \text{degree} \cdot d)$. The core efficiency improvement of NOPE* lies in the iterative coarsening loop. In each iteration, merging nodes $u$ and $v$ into $w$ incurs a feature aggregation cost of $\mathcal{O}(d)$. Furthermore, retrieving the neighbors of node $w$ takes $\mathcal{O}(\Delta)$ time, and for each neighbor, they share the same neighborhood interference term, with the calculation time being $\mathcal{O}(d)$. Therefore, this part consumes a total of $\mathcal{O}(\Delta) + \mathcal{O}(d) \approx O(d)$ time. Thus, the complexity of each iteration is $\Delta \cdot \mathcal{O}(d) = \mathcal{O}(\Delta \cdot d)$. Considering that the algorithm performs $\mathcal{O}(n)$ merge operations, the overall time complexity is approximately $\mathcal{O}(n \cdot \Delta \cdot d)$. Therefore, the total time complexity of NOPE* is $\mathcal{O}(n \cdot \Delta \cdot d)$.

**Space Complexity**: The storage requirements of NOPE* mainly stem from the global feature matrix $\mathcal{X}_{all} \in \mathbb{R}^{n_{max} \times d}$. The graph topology (adjacency list and heap) requires $\mathcal{O}(\mathcal{E})$ space, while the auxiliary size and degree groups consume $\mathcal{O}(n_{max})$. Notably, compared with NOPE*, it avoids the vectorized batch updating process, eliminating the creation of temporary neighbor matrices (which occupy $\mathcal{O}(\Delta \cdot d)$ in NOPE) during runtime. This ensures that the memory footprint remains linear, bounded by $\mathcal{O}(n_{max}d + |\mathcal{E}|)$, minimizing peak memory usage in resource-constrained environments.

## E. Datasets

In this section, we introduce the detailed information of the experimental datasets. And we summarize the statistics in the Table 4.

*Table 4.* Detailed information and statistics of graphs.

| Dataset | Nodes | Edges | Avg degree | Class | Model |
|---------|-------|-------|------------|-------|-------|
| Citeseer | 3,327 | 4,732 | 2.84 | 6 | GNN,LLM |
| Product-subset | 45,855 | 111,638 | 4.87 | 43 | GNN, LLM |
| Ogb-Arxiv | 169,343 | 1,166,243 | 13.77 | 40 | GNN, LLM |
| Book | 594,484 | 3,510,209 | 11.81 | 8 | GNN, LLM |
| Products | 2,449,029 | 123,718,280 | 101.03 | 47 | GNN |

- **Citeseer** (Hu et al., 2020) is a citation network comprising 3,186 scientific publications, where nodes represent individual papers and edges indicate citation relationships. Each node is associated with text attributes derived from the paper's title and abstract.

- **Ogb-Arxiv** (Yang et al., 2016) is a citation network comprising 169,343 Arxiv CS papers and their citation relationships from 40 different academic disciplines. Each node represents a paper, node text attribute is the title and abstract of paper, and each edge represents a citation relationship.

- **Product-subset** (Yang et al., 2016; He et al., 2024; Feng et al., 2024) is a co-purchase graph derived from the Amazon product network, where each node represents a product item and an edge indicates that two products are frequently co-purchased. We adopt the version provided by TAPE (Huang et al., 2024), which is a subset of the original OGBN-Products dataset (He et al., 2024), comprising 45,855 nodes and 111,638 edges. Each node is associated with text attributes such as the product title and description.

- **Book** (Wan & McAuley, 2018) is a literary network from GoodReads, where each node represents a book and the edges represent their similarity relationships. It includes a total of 594,484 books. The node text feature represents the title and description of the book.

- **Products** is the original dataset of Product-subset, which includes 2,449,029 nodes and 123,718,280 edges. The node features of each node are composed of the first 100 final features obtained by PCA.

For all datasets, 10% of the nodes are selected as the testing set. Additionally, in GNN node classification experiments, an extra 50% of the nodes are selected as the training set and 10% as the validation set.

## F. Baseline

Baseline model:

- **FGC** proposes an optimization-driven graph coarsening framework that simultaneously utilizes both the graph structure and node features. By jointly optimizing the coarsened graph and its feature, it theoretically guarantees similarity between the coarsened graph and the original graph within the $\epsilon \in [0, 1)$ range while preserving key graph properties. For the model parameters, where regularization parameters $\lambda = 500, \beta = 0$, relaxation parameters $\alpha = 500, \gamma = 384$.

- **UGC** proposes the Universal Graph Coarsening (UGC) method, which jointly models node attributes and adjacency structures while incorporating dataset dissimilarity factors to adaptively process homophonic and heterophonic graphs. By ensuring spectral similarity and $\epsilon - similarity$, UGC achieves efficient graph reduction while significantly enhancing downstream task performance and computational efficiency. For the model parameters. we select 1,000 hash projectors and each initialized with a uniform norm within the interval [0,1].

- **MPG** proposal a specialized message passing mechanism tailored for coarsened graphs by redesigning the propagation operator on coarsened graphs to enable directed propagation even when the original graph is undirected, while theoretically ensuring signal propagation integrity. In this paper, we set the maximum number of nodes merged at one coarsening step $n_e$ as 100.

- **A-CM** proposes ConvMatch, a graph aggregation method based on convolutional matching, along with its efficient variant A-ConvMatch. By matching and preserving graph convolutional outputs, it directly aligns the convolutional operations of graph neural networks during the aggregation phase, achieving extreme-scale graph coarsening. Regarding

parameter selection, we adhere to the original paper's settings by merging only the top-1 node pairs per round. While this approach impacts model efficiency, it enables precise control over coarsening ratios and delivers optimal results.

Reference model:

- **Random**: For each target node, a neighbor node is randomly selected from its one-hop neighborhood without considering any structural or attribute-related criteria. Subsequently, the neighbor's text is concatenated with the central node's text attributes for evaluation.

- **Degree**: For each target node, select the neighbor node with the highest degree from its one-hop neighborhood. Subsequently, concatenate the neighbor's text with the central node's text attributes for evaluation.

- **RAG**: For each target node, retrieve the semantically most relevant neighbor node from its one-hop neighborhood. Subsequently, concatenate the neighbor text with the central node's textual attributes for evaluation.

- **GCN**: GCN achieves feature propagation and fusion based on graph structure by performing normalized weighted summation of neighbor node features. It leverages Laplacian smoothing to learn node representations, making it well-suited for processing graph data with strong homophony.

- **GIN**: GIN employs learnable summation aggregation functions combined with multilayer perceptrons (MLPs), theoretically possessing discriminative capabilities equivalent to the Weisfeiler–Lehman test. It emphasizes structural discriminability and is suitable for modeling fine-grained structures.

- **GraphSAGE**: GraphSAGE supports inductive learning by sampling neighbors and generating node representations using aggregation functions (mean, pooling, LSTM, etc.), enabling efficient inference on unseen new nodes or new graphs.

- **SGC**: SGC decouples the linear propagation from the nonlinear components in multi-layer GCN, eliminating intermediate nonlinearities and weight matrices. It retains only multi-step feature propagation and linear classifiers, significantly reducing computational complexity while achieving performance comparable to GCN.

## G. Supplementary Prompt Template

### G.1. Prompt Design

---

**The Prompt Template for Supernode Summarization Text**

**Input**: Given a group of {NODE_TYPE}s from the same community within a {GRAPH_TYPE} graph. Each {NODE_TYPE} is represented by its {INFO}.
**Task**: Summarize the content of these {NODE_TYPE}s that are semantically aligned with each other. The summary must explicitly articulate the intersection of their work.
Format requirement: (1) Identify the major recurring themes/directions; (2) Structure the summary around these identified themes; (3) Write the final summary in a cohesive and formal academic style.
**Answer**:

---

**The Prompt Template for Target Node Label Prediction**

**Input**: Given a {GRAPH_TYPE} graph, the target {NODE_TYPE} has the following information: {TARGET_RAW_TEXT}. The target {NODE_TYPE} is related to the following {NODE_TYPE} summary: {SUMMARY_TEXTS}
**Task**: Based on the features of the target {NODE_TYPE} and the {NODE_TYPE} summary, please determine the most appropriate {GRAPH_TYPE} sub-category for the target {NODE_TYPE}.
Categories: {CATEGORY_LIST}.
Please think about the categorization of the target {NODE_TYPE} in a structured manner, and only output the single most relevant category of the target {NODE_TYPE}. Do not give any reasoning or extra text for your answer.
**Answer**:

---

### G.2. Implementation Details

All algorithms are run on a Linux server running Ubuntu 20.04 LTS. All coarsening algorithms are executed on an Intel(R) Xeon(R) Gold 5117 @ 2.0GHz 14C28T (up to 24 cores). Additionally, all evaluation processes are conducted on a GPU (NVIDIA GeForce RTX 4090, 24GB memory) using CUDA 12.6.

## H. Model Performance under Different Coarsening Ratio $r$

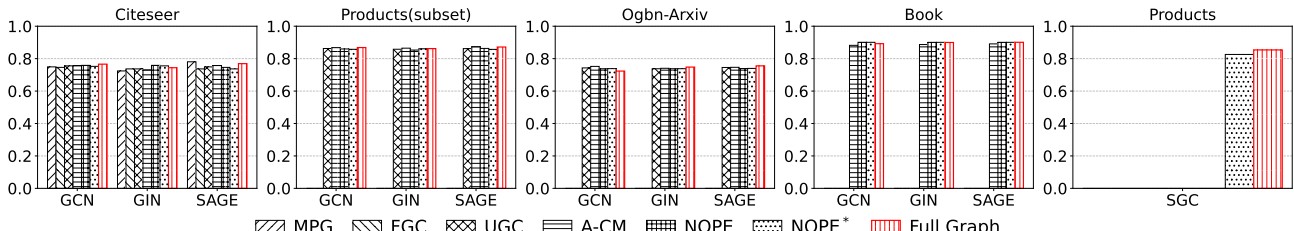

*Figure 8.* Accuracy/Hamming loss for GNN node classification in five datasets under $r = 0.3$.

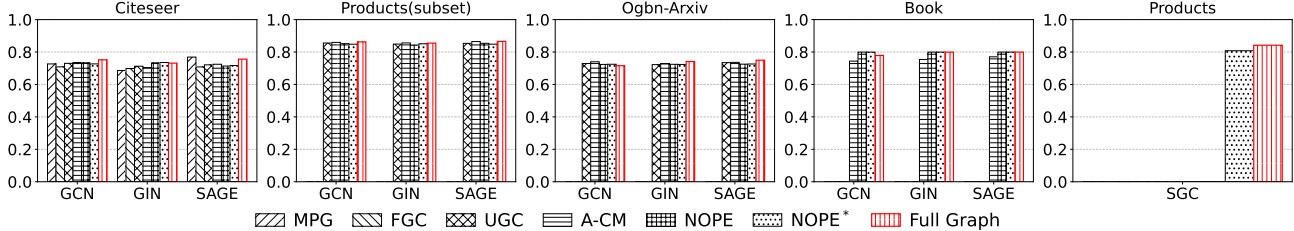

*Figure 9.* F1-score for GNN node classification in five datasets under $r = 0.3$

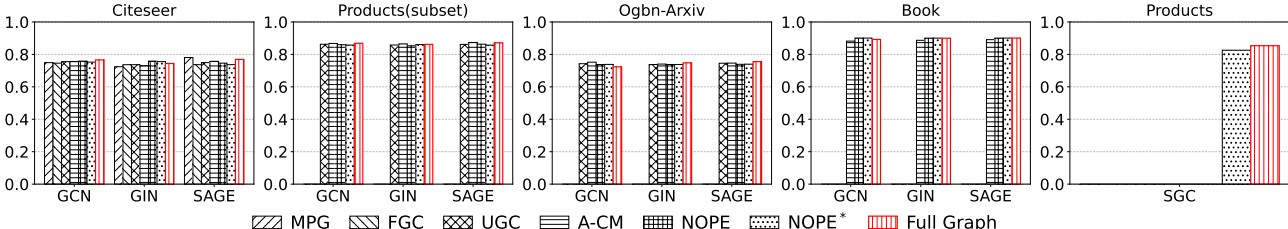

*Figure 10.* Accuracy/Hamming loss for GNN node classification in five datasets under $r = 0.7$.

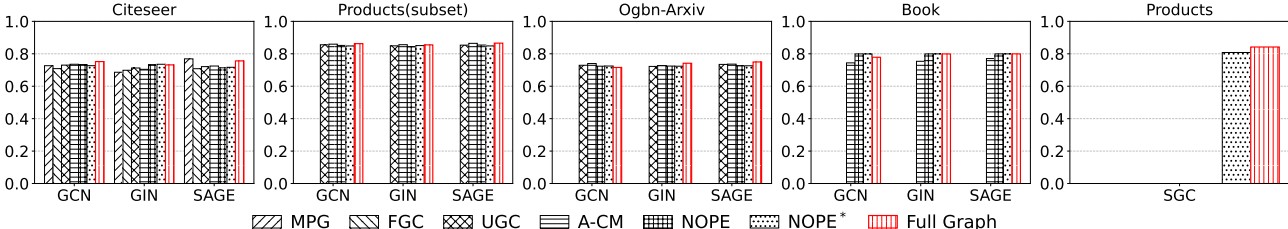

*Figure 11.* F1-score for GNN node classification in five datasets under $r = 0.7$.

In following Tables 5 and 6, to facilitate arrangement, we represent Products(subset) as Products* and abbreviate Ogb-Arxiv as Arxiv.

*Table 5.* The runtime of five datasets under different coarsening ratios.

| Citeseer | 0.1 | 0.2 | 0.3 | 0.4 | 0.5 | 0.6 | 0.7 | 0.8 | 0.9 |
|---|---|---|---|---|---|---|---|---|---|
| MPG | 39.18s | 1m1.30s | 1m14.66s | 1m20.95s | 1m29.07s | 1m37.71s | 1m42.17s | 2m19.27s | 2m54.26s |
| FGC | OOT | OOT | 51m16.78s | 39m23.28s | 35m13.41s | 19m17.19s | 11m11.73s | 5m53.68s | 2m3.97s |
| UGC | 2.26s | 1.73s | 1.67s | 1.43s | 1.30s | 1.12s | 0.99s | 0.86s | 0.49s |
| A-CM | 3.08s | 3.07s | 3.07s | 4.10s | 4.10s | 4.12s | 4.79s | 5.23s | 5.68s |
| NOPE | 0.56s | 0.55s | 0.63s | 0.72s | 0.97s | 0.97s | 1.13s | 1.90s | 2.08s |
| NOPE* | 0.12s | 0.17s | 0.23s | 0.29s | 0.35s | 0.42s | 0.50s | 0.60s | 0.66s |
| Products* | 0.1 | 0.2 | 0.3 | 0.4 | 0.5 | 0.6 | 0.7 | 0.8 | 0.9 |
| MPG | / | / | / | / | / | / | / | / | / |
| FGC | / | / | / | / | / | / | / | / | / |
| UGC | 5m0.35s | 4m26.99s | 3m56.20s | 3m37.92s | 3m1.76s | 2m31.49s | 1m41.46s | 1m1.31s | 39.17s |
| A-CM | 1m0.07s | 1m0.23s | 1m5.35s | 1min5.08s | 1m9.41s | 1m9.52s | 1m24.12s | 1m31.79s | 1m37.62s |
| NOPE | 5.54s | 7.10s | 8.96s | 10.16s | 11.57s | 13.00s | 14.91s | 20.87s | 27.28s |
| NOPE* | 2.52s | 3.48s | 4.45s | 5.49s | 6.46s | 7.71s | 8.76s | 10.17s | 12.43s |
| Arxiv | 0.1 | 0.2 | 0.3 | 0.4 | 0.5 | 0.6 | 0.7 | 0.8 | 0.9 |
| MPG | / | / | / | / | / | / | / | / | / |
| FGC | / | / | / | / | / | / | / | / | / |
| UGC | 59m37.36s | 52m29.46s | 47m2.28s | 41m22.79s | 37m17.51s | 31m30.07s | 20m50.89s | 13m19.27s | 4m23.16s |
| A-CM | 14m30.72s | 14m9.38s | 14m31.37s | 15m39.30s | 16m1.64s | 15m24.22s | 15m50.81s | 16m52.37s | 17m26.59s |
| NOPE | 3m10.36s | 3m37.49s | 4m7.68s | 4m42.78s | 5m12.79s | 5m52.44s | 8m54.41s | 13m1.17s | 44m52.75s |
| NOPE* | 35.04s | 39.44s | 46.80s | 55.24s | 1m4.38s | 1m17.51s | 1m33.45s | 1m54.46s | 2m26.83s |
| Book | 0.1 | 0.2 | 0.3 | 0.4 | 0.5 | 0.6 | 0.7 | 0.8 | 0.9 |
| MPG | / | / | / | / | / | / | / | / | / |
| FGC | / | / | / | / | / | / | / | / | / |
| UGC | / | / | / | / | / | / | / | / | / |
| A-CM | 1h17m11.20s | 1h18m52.25s | 1h13m50.22s | 1h13m44.59s | 1h21m6.50s | 1h16m21.49s | 1h30m35.33s | 1h30m7.56s | 1h35m3.35s |
| NOPE | 8m59.36s | 13min16.56s | 18m5.30s | 28m11.31s | 44m48.80s | 1h22m51.96s | 2h58m32.87s | OOT | OOT |
| NOPE* | 1m54.29s | 2m27.58s | 3m5.46s | 3m50.16s | 4m29.46s | 5m21.88s | 6m34.47s | 8m22.49s | 11m27.32s |
| Products | 0.1 | 0.2 | 0.3 | 0.4 | 0.5 | 0.6 | 0.7 | 0.8 | 0.9 |
| MPG | / | / | / | / | / | / | / | / | / |
| FGC | / | / | / | / | / | / | / | / | / |
| UGC | / | / | / | / | / | / | / | / | / |
| A-CM | / | / | / | / | / | / | / | / | / |
| NOPE | / | / | / | / | / | / | / | / | / |
| NOPE* | 4h42m25.54s | 3h12m36.13s | 2h29m52.64s | 2h3m12.45s | 1h43m9.36s | 1h31m27.73s | 1h22m12.21s | 1h13m21.48s | 1h3m15.73s |

Table 6. The consuming memory of five datasets under different coarsening ratio.

| Citeseer | 0.1 | 0.2 | 0.3 | 0.4 | 0.5 | 0.6 | 0.7 | 0.8 | 0.9 |
|---|---|---|---|---|---|---|---|---|---|
| MPG | 451.79MB | 431.79MB | 468.84MB | 416.36MB | 471.11MB | 417.31MB | 470.61MB | 448.91MB | 412.48MB |
| FGC | 243.12MB | 231.53MB | 203.83MB | 172.72MB | 187.93MB | 193.47MB | 144.30MB | 37.67MB | 49.74MB |
| UGC | 69.90MB | 77.50MB | 72.14MB | 69.13MB | 59.50MB | 46.38MB | 36.38MB | 30.45MB | 25.43MB |
| A-CM | 91.73MB | 90.67MB | 104.58MB | 109.95MB | 96.73MB | 92.68MB | 89.66MB | 114.24MB | 120.79MB |
| NOPE | 12.25MB | 13.12MB | 14.88MB | 15.75MB | 17.50MB | 17.50MB | 19.25MB | 20.12MB | 19.25MB |
| NOPE* | 12.07MB | 13.43MB | 14.30MB | 15.49MB | 16.05MB | 16.78MB | 18.14MB | 19.55MB | 20.28MB |
| Products* | 0.1 | 0.2 | 0.3 | 0.4 | 0.5 | 0.6 | 0.7 | 0.8 | 0.9 |
| MPG | / | / | / | / | / | / | / | / | / |
| FGC | / | / | / | / | / | / | / | / | / |
| UGC | 7,225.50MB | 6,421.10MB | 5,666.80MB | 5,241.80MB | 4,331.60MB | 3,623.00MB | 2,513.60MB | 1,431.00MB | 901.50MB |
| A-CM | 1,302.47MB | 1,261.29MB | 1,291.98MB | 1,225.85MB | 1,228.96MB | 1,075.68MB | 1,164.57MB | 1,102.46MB | 1,205.55MB |
| NOPE | 184.62MB | 199.50MB | 212.62MB | 227.50MB | 240.62MB | 254.62MB | 268.62MB | 281.75MB | 294.00MB |
| NOPE* | 299.14MB | 285.18MB | 269.91MB | 256.79MB | 243.06MB | 229.73MB | 215.54MB | 201.39MB | 186.79MB |
| Arxiv | 0.1 | 0.2 | 0.3 | 0.4 | 0.5 | 0.6 | 0.7 | 0.8 | 0.9 |
| MPG | / | / | / | / | / | / | / | / | / |
| FGC | / | / | / | / | / | / | / | / | / |
| UGC | 88,335.60MB | 77,347.90MB | 69,565.90MB | 60,981.60MB | 53,885.60MB | 46,148.70MB | 30,388.40MB | 19,513.00MB | 6,346.90MB |
| A-CM | 6,580.30MB | 6,553.71MB | 6,572.69MB | 6,498.28MB | 6,526.97MB | 6,488.29MB | 6,536.53MB | 7,555.96MB | 8,528.61MB |
| NOPE | 1,041.51MB | 1,095.64MB | 1,167.07MB | 1,230.91MB | 1,307.60MB | 1,391.21MB | 1,479.27MB | 1,595.02MB | 2,156.32MB |
| NOPE* | 1,109.54MB | 1,164.48MB | 1,225.16MB | 1,287.05MB | 1,354.61MB | 1,425.54MB | 1,515.77MB | 1,590.35MB | 1,642.17MB |
| Book | 0.1 | 0.2 | 0.3 | 0.4 | 0.5 | 0.6 | 0.7 | 0.8 | 0.9 |
| MPG | / | / | / | / | / | / | / | / | / |
| FGC | / | / | / | / | / | / | / | / | / |
| UGC | / | / | / | / | / | / | / | / | / |
| A-CM | 27,333.19MB | 27,357.56MB | 27,362.19MB | 27,290.74MB | 27,351.27MB | 31,317.80MB | 27,276.52MB | 27,283.93MB | 32,553.43MB |
| NOPE | 3,581.41MB | 3,969.21MB | 4,374.80MB | 4,867.13MB | 5,542.74MB | 6,529.45MB | 8,376.42MB | / | / |
| NOPE* | 3,695.39MB | 3,952.46MB | 4,284.29MB | 4,563.70MB | 4,837.80MB | 5,087.12MB | 5,279.05MB | 5,453.23MB | 5,628.05MB |
| Citeseer | 0.1 | 0.2 | 0.3 | 0.4 | 0.5 | 0.6 | 0.7 | 0.8 | 0.9 |
| MPG | / | / | / | / | / | / | / | / | / |
| FGC | / | / | / | / | / | / | / | / | / |
| UGC | / | / | / | / | / | / | / | / | / |
| A-CM | / | / | / | / | / | / | / | / | / |
| NOPE | / | / | / | / | / | / | / | / | / |
| NOPE* | 39,288.35MB | 39,131.04MB | 38,455.35MB | 37,070.87MB | 35,348.83MB | 33,603.52MB | 31,926.63MB | 30,209.79MB | 28,408.45MB |

Table 7. LLM node classification results on different datasets under $r = 0.3$.

| Node Classification | | Full Graph | | | Graph Coarsening | | | | Ours | |
|---|---|---|---|---|---|---|---|---|---|---|
| **Dataset** | **Metric** | *Random* | *Degree* | *RAG* | FGC | MPG | UGC | A-CM | NOPE | NOPE* |
| Citeseer | ACC | *0.5768* | *0.5924* | *0.5956* | 0.5924 | 0.6081 | 0.5799 | 0.6050 | 0.6207 | 0.5956 |
| | F1 | *0.5972* | *0.6098* | *0.6147* | 0.6093 | 0.6247 | 0.5944 | 0.6195 | 0.6366 | 0.6107 |
| Products(subset) | ACC | *0.5859* | *0.5761* | *0.5789* | / | / | 0.6037 | 0.6690 | 0.6737 | 0.6796 |
| | F1 | *0.6078* | *0.5980* | *0.6003* | / | / | 0.6156 | 0.6698 | 0.6779 | 0.6830 |
| Ogb-Arxiv | ACC | *0.4142* | *0.4029* | *0.4256* | / | / | 0.3475 | 0.3829 | 0.3746 | 0.3700 |
| | F1 | *0.4046* | *0.3940* | *0.4154* | / | / | 0.3432 | 0.3829 | 0.3746 | 0.3653 |
| Book | ACC | *0.8994* | *0.8996* | *0.9080* | / | / | / | 0.9184 | 0.9200 | 0.9240 |
| | F1 | / | / | / | / | / | / | / | / | / |

*Table 8.* LLM node classification results on different datasets under $r = 0.7$.

| Node Classifiction | | Full Graph | | | Graph Coarsening | | | | Ours | |
|---|---|---|---|---|---|---|---|---|---|---|
| **Dataset** | **Metric** | *Random* | *Degree* | *RAG* | FGC | MPG | UGC | A-CM | NOPE | NOPE* |
| Citeseer | ACC | 0.5768 | 0.5924 | 0.5956 | 0.5817 | 0.6050 | 0.5517 | 0.6269 | 0.5987 | 0.6175 |
| | F1 | 0.5972 | 0.6098 | 0.6147 | 0.5963 | 0.6222 | 0.5711 | 0.6396 | 0.6150 | 0.6326 |
| Products(subset) | ACC | 0.5859 | 0.5761 | 0.5789 | / | / | 0.5187 | 0.6768 | 0.6842 | 0.6881 |
| | F1 | 0.6078 | 0.5980 | 0.6003 | / | / | 0.5420 | 0.6780 | 0.6860 | 0.6917 |
| Ogb-Arxiv | ACC | 0.4142 | 0.4029 | 0.4256 | / | / | 0.3200 | 0.4105 | 0.3880 | 0.3937 |
| | F1 | 0.4046 | 0.3940 | 0.4154 | / | / | 0.3121 | 0.3965 | 0.3785 | 0.3862 |
| Book | ACC | 0.8994 | 0.8996 | 0.9080 | / | / | / | 0.9140 | 0.9104 | 0.9255 |
| | F1 | / | / | / | / | / | / | / | / | / |

