# OpenReview forum: "Rethinking Efficient Graph Coarsening via a Non-Selfishness Principle"
_ICML.cc/2026/Conference — ICML 2026 regular_

### Official Review · Reviewer_YxeE · 2026-03-11

**Soundness:** 2
**Presentation:** 3
**Significance:** 3
**Originality:** 2
**Overall Recommendation:** 4
**Confidence:** 3

**Summary:**

This paper studies graph coarsening for text-attributed graphs and argues that many existing approaches rely on selfish pairwise matching that ignores the effect of a merge on the surrounding neighborhood. The authors propose a neighborhood interference index, Equation (1), and build a greedy coarsening algorithm, NOPE, that uses this index to prioritize merges with low neighborhood disturbance. They further introduce NOPE*, an approximation based on a local isotropy assumption, Equation (2), to reduce the merge evaluation cost from degree-dependent to feature-dimension-dependent. Experiments on several graph datasets evaluate runtime, memory, and downstream node classification with both GNNs and LLM-based prompting, with the main finding being that NOPE* is much faster than prior coarsening baselines while remaining broadly competitive on downstream tasks.

**Compliance With Llm Reviewing Policy:**

Affirmed.

**Final Justification:**

My major concerns were addressed.

**Key Questions For Authors:**

See weaknesses

**Limitations:**

Yes

**Strengths And Weaknesses:**

**Strengths:**

1. The most convincing aspect of the paper is the runtime and memory story. Table 1 on Page 6 is genuinely useful because it includes both small and large graphs and reports failure modes such as OOM and OOT. Figures 3 and 4 reinforce that the advantage is not confined to a single coarsening ratio.
2. The formulation in Equation (1) on Page 3 is easy to understand, and the algorithmic design with cached statistics plus a min-heap is practical.
3. Figure 2 on Page 5 is one of the better parts of the paper. It shows a concrete failure mode of NOPE*, and it directly explains why aggressive coarsening can become riskier.
4. The paper tries to connect coarsening to both GNNs and LLM-style graph reasoning.

**Weaknesses:**

1. The core novelty is weaker than claimed. The paper defines a shift from “selfishness” to “non-selfishness” as if prior coarsening methods only optimize naive pairwise affinity. That framing is too clean. Some cited baselines, especially MPG and A-CM, already preserve broader propagation or convolutional behavior, not just local pair similarity. So the main contribution is better described as a new local merge criterion and an approximation scheme, not a major conceptual novelty.
2. The motivational evidence for the non-selfishness principle is too narrow. Figure 1 compares the proposed principle against a handcrafted selfish baseline on Citeseer using Dirichlet Energy and Avg. EBC. The trend is interesting, but one toy comparison on one dataset is not enough to support the narrative that prior methods are fundamentally flawed because they are selfish. If the principle is central, it should be tested more directly against actual competitive baselines and on more than one graph.
3. Assumption 4.1 and Equation (2) are the main component of the fast method. But the paper never verifies whether local isotropy is a reasonable approximation for the Sentence-BERT embeddings on these graphs. Worse, Figure 2 already shows that the approximation degrades as coarsening progresses. Without an empirical isotropy diagnostic, the theory feels detached from the actual data regime in which the method is used.
4. The experimental analysis lacks uncertainty estimates and exact reporting where they are needed most.
5. There are several clarity and consistency issues.

---

> ### Author Rebuttal · Authors · 2026-03-31
>
> W1: Thank you very much for your suggestion. The most significant shift lies not in the method used to calculate similarity, but rather in whether one explicitly models and optimizes the impact of a merge on the entire neighborhood. This approach enables us to directly control the neighborhood-level distortion induced by the merging process. Previous methods demonstrated their sophistication primarily through various techniques for calculating similarity, yet they failed to quantify the cost incurred by the neighborhood as a result of such sophisticated merging—and this represents precisely the conceptual shift we have made, moving from a "non-self" perspective to a "selfish" one.
>
> W2: Thank you for your comments. To address this concern, we have supplemented our results with experiments on two additional datasets(https://anonymous.4open.science/r/NOPE-FA74/Rebuttal/review4/citeseer.png and https://anonymous.4open.science/r/NOPE-FA74/Rebuttal/review4/products(subset).png); It shows that our methods achieve competitive performance overall. While they do not consistently outperform all baselines in terms of Dirichlet Energy, they maintain a well-balanced level of smoothness—avoiding both excessive over-smoothing and unstable high-variation regimes. Importantly, this is complemented by consistently strong performance in Avg. EBC, indicating that our approach better preserves structural balance during coarsening.
> More generally, Dirichlet Energy and Avg. EBC  should be viewed as motivational evidence, while the main empirical support comes from the broader comparisons against competitive baselines across multiple datasets.
>
> W3: Thank you for this important comment. Prior work, such as 'On the Sentence Embeddings from Pre-trained Language Models', shows that BERT-generated embeddings exhibit a certain degree of anisotropy, which provides support for our assumption and the design of NOPE\*. In addition, existing methods such as anisotropy regularization and IsoScore have been proposed to characterize anisotropy, and we will further incorporate this related literature in the revised paper.
> Moreover, the strong agreement between the interference curves of NOPE and NOPE\* in Figs. 2 and 7 during the early and middle stages of the coarsening process also provides empirical support for the reasonableness of this approximation. At the same time, we acknowledge that the divergence between the two becomes more pronounced under very high coarsening ratios. This remains an open issue, and for this reason, we do not recommend using excessively high coarsening ratios in practice.
>
> W4: Thank you very much for your suggestion. In the LLM-based node classification experiments, the temperature was set to 0, so the outputs were deterministic across runs. In contrast, the GNN-based results in this paper are averaged over multiple runs under five seeds and report the mean ± standard deviation, and the key improvements have been shown to be statistically significant following statistical testing.
> To provide a more complete view of the experimental results, we have also included the original classification results for all runs; please refer to (https://anonymous.4open.science/r/NOPE-FA74/Rebuttal/review4/Weakness4_variance.png).
>
> W5: Thank you for pointing this out. We have indeed identified several instances of unclear descriptions and ambiguous references (including some unclear referents and symbols), and we will address these in a subsequent revision.

---

> > ### Author Rebuttal · Reviewer_YxeE · 2026-04-03
> >
> > Thank you for the detailed rebuttal. It has improved my assessment of the paper, and I am increasing my score by one point.
> >
> > The rebuttal helps clarify that the main contribution is the explicit modeling of neighborhood-level merge interference, rather than simply another similarity function, which improves the positioning of the work. I also appreciate the added discussion and supplementary evidence on the non-selfishness motivation, as well as the clarification around uncertainty reporting and significance testing.
> >
> > The response on the NOPE* approximation is also helpful. While I still think an explicit isotropy diagnostic on the actual embeddings would strengthen the paper further, the authors have now better acknowledged the approximation regime and its limitations at high coarsening ratios. My concerns are only partially resolved, but the rebuttal has positively changed my view of the submission.

---

> > > ### Author Response · Authors · 2026-04-04
> > >
> > > Thank you for your thoughtful feedback and for reconsidering your assessment of the paper. We truly appreciate the opportunity to clarify the contributions and motivations behind our work.
> > > Regarding your concerns about the NOPE\* approximation method, we agree that explicitly diagnosing the isotropy of the actual embedding vectors would indeed provide additional value. Although our primary objective was to justify our experimental setup from a performance perspective, we greatly appreciate your suggestion and will certainly give it serious consideration in our future work.
> > > Once again, thank you for your constructive feedback. It has been invaluable in refining our understanding and presentation of the work.

---

### Official Review · Reviewer_GVNa · 2026-03-12

**Soundness:** 2
**Presentation:** 2
**Significance:** 2
**Originality:** 2
**Overall Recommendation:** 3
**Confidence:** 4

**Summary:**

The paper introduces a method of graph coarsening aiming to merge nodes based on the aggregate change of similarity with the neighbouring of a given node pair; authors later introduce a computationally efficient relaxation on the method based on the feature isotropy.
To illustrate the method performance, authors estimate its computational complexity and use a variety of GNN architectures to generate embeddings of the coarsened supernodes.
These embeddings are appended to the corresponding node features and fed into the MLP performing node classification; the experiments demonstrate computational efficiency of the suggested coarsening routine and non-deteriorating performance for the node classification tasks.

**Compliance With Llm Reviewing Policy:**

Affirmed.

**Final Justification:**

As mention in my rebuttal acknowledgement, I am not completely convinced by the method. The speed is improved, yes. But the method still seems to be ad-hoc, and the performance improvements marginal.

**Key Questions For Authors:**

See Weaknesses #1, #3, #4, #5.

**Limitations:**

yes

**Strengths And Weaknesses:**

**Strengths**: The paper is based on the well-motivated insight that graph coarsening should account for more than pairwise feature similarity for the node merging; specifically, one may aim to preserve the local topology of the similarities (to which authors refer as "non-selfishness").
Whilst in principle such approach may be computationally demanding, the suggested NOPE method is claimed to be computationally effective.

**Weaknesses**:

1. How does one compute the index $I$ using $\Sigma$? It is not explained in the text, and I believe that the attached code actually computes a different value: $I_{pq} = \sum_{i \in N_p} s_{ip}^2 +  \sum_{i \in N_p} s_{ip}^2 + \sum_{i \in N_p \cap N_q} (s_{ip}-s_{iq})^2$, which does not equal to the original definition (unless $s_{ij}=0$ if there is no corresponding edge which is not stated...). Simple example of the problem -- compute $I_{23}$ for the graph (1) -- (2) -- (3). Additionally, the computational cost of $I$ should be adjusted accordingly (although asymptotic behaviour may remain the same) and properly explained in the appendix.
2. How realistic is isotropic assumption? Isn't it basically assume white noise as features? Moreover, the expected index $I$ in NOPE\* is namely the pairwise similarity weighted by the product of degrees: is it a desired behaviour? For instance, it implies that a pair of very similar nodes may never be merged (for a chosen $r$) if both of them are connected to a large number of dissimilar nodes.
3. Note that by its definition merging is a smoothing operation. In the experimental setting features are iteratively smoothed during the coarsening, and then are fed into various GNN architectures which are also prone to oversmoothing by themselves. One can argue that this is the reason why there is no big variation in classification results: new embeddings are uniformly oversmoothed. As a side node, I would suggest a baseline where just a random vector of features is appended to MLP to see the performance change.
4. The value of $r$ seems quite low (see, f.i. A-CM paper, where they use $r=0.9$ and $r=0.99$): $r=0.5$ does not seem to be a sufficient reduction of the dimensionality. Given the Table for $r=0.7$, it seems at least possible that NOPE's computation may be more demanding which is a problem, since the computational efficiency is key gain of the algorithm.
5. The GNN-classification architecture is questionable. Note that there was quite a number of attempts of appending various versions of adjacency lists as features to the MLP input; so in principle this is not a novel attempt. The evaluation and comparison conducted in A-CM paper would be extremely beneficial for the current work; accounting for the bullet points 3 and 4 above, the current level of results does not seem sufficient.

Further the authors discuss that the approach loses efficiency for denser graphs, however a further proper discussion of the limitations induced by higher $r$ and oversmoothing is required.

---

> ### Author Rebuttal · Authors · 2026-03-31
>
> W1: Thank you for your suggestion. In our implementation, if a node i belongs only to the neighborhood of p, we set $s_{iq}$=0 (and symmetrically for q). Under this masking convention, the implemented form is equivalent to the $I_{pq}$ defined in the paper. We adopt this convention because otherwise the merged similarity would collapse toward $s_{iw}$, making it difficult to capture the information difference introduced by the merge. We will make this convention explicit in the revision and add a toy example in the appendix to illustrate the computation step by step.
> We will also clarify that $\Sigma$ is used to cache neighborhood-related computations and support incremental updates after merging, which improves efficiency without changing the overall asymptotic complexity.
>
> W2: Thank you for your suggestion. We would like to clarify that the local isotropy assumption in NOPE\* should not be interpreted as assuming that node features are white noise or globally structureless. Rather, this assumption is closely related to the node feature generation process. Prior work, such as 'On the Sentence Embeddings from Pre-trained Language Models', has shown that BERT-generated embeddings exhibit a certain degree of anisotropy, which provides empirical support for our assumption and the motivation behind NOPE\*. This interpretation is also consistent with our empirical observations: the close fit between the NOPE and NOPE\* interference curves in the early and middle stages suggests that isotropy can serve as a useful local approximation in the low-to-moderate coarsening regime.
> Regarding the expected index in NOPE\*, we consider this a reasonable derivation rather than an artificial flaw. Nodes that appear similar but are embedded in different neighborhood contexts may be similar only in content, but not in structural role.  NOPE\* explicitly constrains this through the compositional relationships within the neighborhood.
>
> W3: Thank you for your suggestion. Oversmoothing is an inherent issue in the coarsening process. To examine this effect, we use Dirichlet energy as a reference metric. The results(Fig 1 in paper and Figs in https://anonymous.4open.science/r/NOPE-FA74/Rebuttal/review3/citeseer.png and https://anonymous.4open.science/r/NOPE-FA74/Rebuttal/review3/products(subset).png) show that, compared with other coarsening methods, NOPE can alleviate smoothing to some extent.
> Following your suggestion, we  added an experiment: replacing the output features of GNN with random vectors. The result (https://anonymous.4open.science/r/NOPE-FA74/Rebuttal/review3/Weakness3_random_vector.png) shows that replacing the GNN output with random vectors leads to a significant performance drop, indicating that the GNN output contributes substantially to the final results and still preserves considerable information on the coarsened graph.
>
> W4: Thank you for your suggestion. The coarsening ratios used in this paper follow the settings adopted in the FGC and UGC papers. Hence, we report results at r=0.3, 0.5, and 0.7. In fact, we recommend using NOPE\* to avoid node height limitations under ultra-high coarsening rates. Additonally, we supplement experiments at more aggressive coarsening ratios, r= 0.01. The running time and running memory results (https://anonymous.4open.science/r/NOPE-FA74/Rebuttal/review3/Weakness4%EF%BC%882%EF%BC%89_r=0.01_running_info.png) and the node classification results (https://anonymous.4open.science/r/NOPE-FA74/Rebuttal/review3/Weakness4%EF%BC%881%EF%BC%89_r=0.1:0.01.png) show that the performance degrades under these settings.
>
> W5: Thank you for your suggestions. In fact, the GNN-based classification architecture itself is not a methodological contribution of this paper. Our intention is not to claim novelty in concatenating node features with graph-based representations, but rather to adopt a unified downstream evaluation protocol for all coarsening methods under comparison.
> In addition, we conducted some additional experiments under the A-CM setting, and the corresponding results are reported in the table（https://anonymous.4open.science/r/NOPE-FA74/Rebuttal/review3/Weakness5_new_node_classification.png）. The results show that our method still achieves strong performance. Compared with A-CM and UGC, which appear more prone to overfitting(possibly because A-CM/UGC are more affected by the BERT-generated embeddings), NOPE demonstrates stronger robustness to overfitting.
>
> Last Review: Thank you for your suggestions. To address the efficiency issue, we proposed NOPE* and recommended its use at a coarsening rate; indeed, this approach demonstrated a significant improvement in efficiency. Furthermore, regarding the issue of over-smoothing, it is undeniable that the algorithm does, to some extent, exhibit over-smoothing—particularly at high coarsening rates (r=0.1 and 0.01); however, under the experimental settings adopted in this paper, this over-smoothing remains controllable.

---

> > ### Author Rebuttal · Reviewer_GVNa · 2026-04-02
> >
> > I thank the authors for their responses. These address some of my concerns, even though I am admittedly still not completely convinced by the presented approach. In terms of speed, there appears to be a gain compared to other methods, in terms of performance not so much. I would be willing to increase my score by 1, but I am still more leaning towards a weak rejection.

---

> > > ### Author Response · Authors · 2026-04-04
> > >
> > > Thank you very much for your valuable suggestions and guidance regarding our work.
> > > First, regarding the time and space efficiency of the algorithm, the model has indeed achieved significant progress, and we appreciate your acknowledgment of this. In terms of model performance, we adhered to the experimental settings you advised—specifically the A-CM/UGC setting(i.e., training on the coarsening graph with edge weights and testing on the original graph). Regarding the specific experimental results, we conducted a comprehensive comparison of all algorithms. A selection of the experimental results is presented below(The complete set of experimental results can be viewed at https://anonymous.4open.science/r/NOPE-FA74/Rebuttal/review3/new_gnn_all1.png):
> > >
> > > Citeseer (r=0.5)
> > > | Metric         | FGC     | UGC     | A-CM    | NOEP   | NOPE*  |
> > > |----------------|---------|---------|---------|--------|--------|
> > > | **ACC**        | 0.7304  | 0.7304  | 0.7524  | 0.79   | 0.8025 |
> > > | **F1-weighted**| 0.6934  | 0.6917  | 0.7178  | 0.7752 | 0.7947 |
> > >
> > >
> > >
> > > Products (Subset) (r=0.5)
> > > | Metric         | FGC     | UGC     | A-CM    | NOEP   | NOPE*  |
> > > |----------------|---------|---------|---------|--------|--------|
> > > | **ACC**        | /       | 0.8471  | 0.8753  | 0.8766 | 0.8925 |
> > > | **F1-weighted**| /       | 0.8372  | 0.8672  | 0.869  | 0.8856 |
> > >
> > >
> > > Book (r=0.5)
> > > | Metric         | FGC     | UGC     | A-CM    | NOEP   | NOPE*  |
> > > |----------------|---------|---------|---------|--------|--------|
> > > | **Ham Loss**   | /       | /       | 0.1364  | 0.1334 | 0.1372 |
> > > | **F1**         | /       | /       | 0.7239  | 0.7328 | 0.7101 |
> > >
> > > The experimental results demonstrate that, in terms of GNN-based node classification performance under the new experimental settings, our algorithm exhibits exceptional competitiveness. Furthermore, given the robust performance the algorithm demonstrated when processing LLM-based node classification task—coupled with its outstanding time and space efficiency—we are confident that its overall performance holds immense promise. Moving forward, we intend to further refine our experimental settings and report the latest experimental results.

---

### Official Review · Reviewer_g4zr · 2026-03-12

**Soundness:** 2
**Presentation:** 2
**Significance:** 3
**Originality:** 3
**Overall Recommendation:** 3
**Confidence:** 4

**Summary:**

This paper studies text-attributed graph coarsening and proposes a new “non-selfishness” principle for merge decisions. Instead of selecting merge pairs solely based on pairwise similarity, the paper defines a neighborhood interference index  to measure how much a candidate merge perturbs the surrounding neighborhood semantics. Based on this metric, the authors propose NOPE, a greedy interference-aware coarsening algorithm with cached updates, and a faster approximation NOPE* that replaces the exact interference with an expectation under a local isotropy assumption. The paper claims near-linear time and linear memory for NOPE, and further reduces the per-merge evaluation cost. Experimentally, the paper shows substantial runtime and memory improvements on several datasets.

**Compliance With Llm Reviewing Policy:**

Affirmed.

**Final Justification:**

I thank the authors for the clarification and additional experiments, which are helpful. However, my concern in W2 is still not fully addressed. The response mainly reframes Dirichlet energy as an auxiliary diagnostic, without providing sufficiently theoretical evidence that the method mitigates oversmoothing or preserves feature diversity. Therefore, I will maintain my original score.

**Key Questions For Authors:**

Q1. Can you explain whether the previous methods neglect the adjacent information?
Q2. How do you design the selfish method in Section 4.1?
Q3. How do you derive the difference claim between Graph Condensation and Feature Graph Coarsening?

**Limitations:**

Yes

**Strengths And Weaknesses:**

S1. The main contribution of efficiency and scalability is validated by the theoretical and experimental results. NOPE maintains caches and performs greedy contractions with incremental updates, while NOPE* further replaces the exact interference score by an approximate surrogate derived under local isotropy. The claimed complexity reduction is meaningful, where NOPE and NOPE* all both keep linear-space behavior.

S2. The empirical efficiency results seem quite impressive. For each dataset, this paper proposes two evaluation frameworks to assess the effectiveness of the coarsening results. With LLM-based and GNN-based frameworks, NOPE shows a practical gain in respect of efficiency compared with other baselines, especially on the large graph.

S3. The paper is reasonably technical in showing that NOPE* is a speed-oriented approximation rather than claiming it is equivalent to NOPE. In fact, the paper shows that the interference process of NOPE* gradually diverges from NOPE as coarsening proceeds, and explicitly states that the approximation is most suitable at low-to-moderate coarsening rates. This honesty is a positive aspect of the presentation.

W1. My main concern is that the motivation is overclaimed. The paper repeatedly mentions previous graph coarsening methods as “pair-wise” and “selfish”, claiming that they ignore neighborhood information. However, this is not convincing. Even in the paper’s own related work section, methods such as UGC are described as using node attributes and adjacency information, and MPG / A-CM / CoCoA are described as designing criteria aligned with message propagation or GNN-specific properties. Therefore, the claim that prior methods generally neglect neighborhood information is not sufficiently convincing. The novelty would be better positioned as introducing an explicit neighborhood-merge objective, rather than as the motivation of pairwise reasoning.

W2. My second concern is about Section 4.1 and the comparison of the Dirichlet Energy. The paper argues that higher Dirichlet Energy under the proposed method shows better semantic preservation and less over-smoothing. This conclusion can not convince me. Higher Dirichlet Energy does not necessarily show better semantics, and it may also mean insufficient smoothing or fewer communities. More importantly, this comparison is not performed against the paper’s main baselines, such as FGC, UGC, MPG, or A-CM, but against a defined “selfish” method, which is not explained in detail. Thus, the evidence is too weak to support the strong claims about existing methods.

W3. The presentation of this paper can be further improved. For example, the abstract mentions a large improvement with respect to the complexity, but it never explains the concrete meaning of each signal, such as d and delta. Another example is that the presentation in related work always confuses me:  “... making them unsuitable for our setting”. What is the setting of this paper? Why are they unsuitable? Finally, the difference claim between Graph Condensation and Feature Graph Coarsening is too casual, as it has no reference or theoretical proof.

W4. Fourth, the complexity claims should be phrased more precisely. The key improvement of NOPE* is reducing the dependence on the dynamic degree from quadratic to linear. That is a real contribution. However, calling the method “near-linear” may be somewhat too flattering unless one emphasizes that this relies on the practical sparsity assumption Δ≪n, rather than being truly linear in all relevant situations.

---

> ### Author Rebuttal · Authors · 2026-03-31
>
> W1: Thank you for your feedback. We agree that prior graph coarsening methods do incorporate neighborhood information, and our intention was not to suggest otherwise.
> Our distinction is not about whether neighborhood information is present in node features, but about the objective guiding the merge. Existing methods are fundamentally pair-centric: they prioritize merging similar nodes, with neighborhood effects only implicitly reflected through representations. In this sense, their objectives remain "selfish" as they focus on local pairwise similarity.
> In contrast, our notion of "non-selfish" lies in the merge objective itself. Rather than only selecting similar node pairs, we explicitly model and optimize the impact of a merge on the entire neighborhood. This allows us to directly control the neighborhood-level distortion induced by merging, instead of relying on it being indirectly captured by node features.
>
> W2: Thank you for your feedback. Indeed, oversmoothing is an inherent challenge in graph coarsening('A Comprehensive Survey on Graph Reduction: Sparsification, Coarsening, and Condensation', 'Graph coarsening: from scientific computing to machine learning'). To assess whether our strategy mitigates this effect and preserves feature diversity, we employ Dirichlet energy as an auxiliary diagnostic measure. Actually, in our work, Dirichlet energy serves primarily as a supporting signal rather than a definitive criterion. The primary evaluation of our method is still based on its performance in downstream tasks.
> We also conducted additional experiments on Citeseer and Products(subset), comparing NOPE/NOPE\* with multiple baselines across different coarsening ratios. The visualized results can be found in https://anonymous.4open.science/r/NOPE-FA74/Rebuttal/review2/citeseer.png and https://anonymous.4open.science/r/NOPE-FA74/Rebuttal/review2/products(subset).png. It shows that our methods achieve competitive performance overall. While they do not consistently outperform all baselines in terms of Dirichlet Energy, they maintain a well-balanced level of smoothness—avoiding both excessive over-smoothing and unstable high-variation regimes. Importantly, this is complemented by consistently strong performance in Avg. EBC, indicating that our approach better preserves structural balance during coarsening.
>
> W3: Thank you for the valuable suggestions. First, we acknowledge that key variables in the complexity analysis are not clearly introduced. Although defined in Table 3, symbols such as d (feature dimension) and $\Delta$ (dynamic average degree) are not explained at their first occurrence. We will revise the paper to define all key variables more explicitly for improved clarity.
> Second, Our setting focuses on feature graph coarsening for text-attributed graphs, where both node features and structure are essential. Some traditional methods rely primarily on structural information, which limits their applicability. We will clarify this point more precisely.
> Finally, We will revise this part to provide a clearer and better-supported distinction—highlighting that coarsening aggregates real nodes to preserve graph properties, whereas condensation learns a compact synthetic graph for downstream tasks—and include appropriate references (e.g., 'A Comprehensive Survey on Graph Reduction: Sparsification, Coarsening, and Condensation' and 'Graph Condensation: A Survey').
>
> W4: Thank you for the valuable suggestions. Our use of near-linear refers to the practical regime where the dynamic degree $\Delta$ remains relatively small during the coarsening process ($\Delta$ << n). Under this condition, the overall complexity can be reasonably approximated as linear in n.
> This behavior is most evident in sparse graphs, where degree growth during merging is naturally limited. When the coarsenion process does not significantly increase node degrees, the algorithm exhibits scaling close to linear. In cases of aggressive coarsening or graphs with dense local structures, $\Delta$ may grow substantially, leading to super-linear behavior. Therefore, NOPE is not strictly linear in general, but approaches linear scaling when \delta remains controlled—particularly in sparse settings.

---

> > ### Author Rebuttal · Reviewer_g4zr · 2026-04-03
> >
> > Thank you for the clarification and additional experiments, which are helpful. However, my concern in W2 is still not fully addressed. The response mainly reframes Dirichlet energy as an auxiliary diagnostic, without providing sufficiently theoretical evidence that the method mitigates oversmoothing or preserves feature diversity. Therefore, I will maintain my original score.

---

> > > ### Author Response · Authors · 2026-04-04
> > >
> > > Thank you for your insightful comments. In fact, several prior studies have already elucidated the relationship between Dirichlet energy and smoothing(feature diversity)—notably, "Measuring Over-smoothing beyond Dirichlet energy" and "Dirichlet Energy Enhancement of Graph Neural Networks by Framelet Augmentation". On one hand, these works theoretically demonstrate—from the perspective of spectral analysis—the correlation between the decay of Dirichlet energy and the phenomenon of over-smoothing. On the other hand, they highlight that addressing the over-smoothing problem effectively amounts to preserving feature diversity to a significant extent. This provides us with a theoretical foundation; accordingly, we will revise the descriptions within the paper to enhance the rationale.
> > >
> > > Furthermore, to more rigorously validate this preference for preserving diversity, we additionally designed an experiment where we calculated the mean Dirichlet energy for connected supernodes, distinguishing between those that share the same label (determined by the mode of the labels within all constituent nodes of the supernode, as setting in common graph coarsening model, such as FGC, UGC and A-CM) and those with different labels. This allowed us to assess whether this metric can effectively differentiate between nodes based on the presence or absence of shared labels.
> > > The results are as follows:
> > > |                                      | NOPE   | NOPE*   | UGC     | A-CM   | FGC     | MPG    |
> > > | ------------------------------------ | ------ | ------- | ------- | ------ | ------- | ------ |
> > > | Same Label Dirichlet Energy      | 755.46 | 942.62  | 1047.26 | 672.60 | 963.56  | 586.96 |
> > > | Different Label Dirichlet Energy | 906.44 | 1147.55 | 952.17  | 809.03 | 1140.02 | 698.81 |
> > >
> > > The Dirichlet energy between nodes with the same label is significantly smaller than that between nodes with different labels. These results indicate that, in the context of graph coarsening, the process does not indiscriminately reduce feature diversity. Rather, it applies smoothing operations in a more targeted and controlled manner. Moving forward, we will revise the corresponding experiments to make the overall explanation more convincing.

---

### Official Review · Reviewer_udUQ · 2026-03-14

**Soundness:** 3
**Presentation:** 3
**Significance:** 2
**Originality:** 2
**Overall Recommendation:** 4
**Confidence:** 4

**Summary:**

The authors consider the graph coarsening problem for large text-attributed graphs. A key observation is that when coarsening on graphs, merged nodes should consider their neighbors instead of pairwise similarity. The authors propose the *non-selfishness* way to capture this observation. The paper then proposes NOPE and its faster variant NOPE*. It uses an interference-aware greedy merging rule and NOPE* further approximates the interference score under a local isotropy assumption to reduce time complexity. The experiments on several benchmark datasets show very large gains in runtime and memory.

**Compliance With Llm Reviewing Policy:**

Affirmed.

**Ethical Review Flag:**

Flag this paper for an ethics review.

**Key Questions For Authors:**

Please consider the concerns raised in my weak points. Also, I have the following questions:

- 1. The main approximation in NOPE* depends on the local isotropy assumption. How realistic is this assumption on the datasets used here, and is there any empirical way to measure when it is a good approximation?

- 2. Since the paper shows that approximation bias grows as coarsening proceeds, have the authors considered a hybrid strategy, e.g., using NOPE* in early stages and switching to NOPE later? That seems potentially more robust than using one method throughout. This can be a tradeoff between bias and efficiency.

- 3. In my opinion, the graph coarsening problem is to consider the situation that graph is huge-scale. Can the authors test their algorithms on even larger graphs not just small or medium scale graphs?

**Limitations:**

Yes

**Strengths And Weaknesses:**

Overall, I think the paper has a decent contribution on graph corasening problem. I summarize the strong points and weak points as follows:

- 1. The problem formulation of non-selfishness is interesting and novel. The authors' reframe graph coarsening from pairwise similarity matching to neighborhood-aware interference minimization is a meaningful perspective shift.

- 2. Compared with existing baseline methods, the proposed method is quite efficient. The results (in Table 1) indicate that both proposed NOPE and NOPE* are consistently much faster and more memory-efficient than prior baselines.

- 3. The empirical evaluation is extensive. The paper includes runtime/memory comparisons, GNN-based evaluation, and LLM-based evaluation.

Weak points:

- 1. It seems to me that the time complexity and inference analysis is based on a strong assumption. For example, the quality of NOPE* relies on the local isotropy assumption, but it feels quite strong, and the paper does not really explain when it should be expected to hold in practice.

- 2. The paper claims near-linear behavior, but the actual complexity of NOPE is still $\mathcal{O}(n\Delta^2 d)$, so the practical story is much stronger than the theoretical one. I think the presentation should be a bit more careful here, especially since $\Delta$ can grow during coarsening.

- 3. The efficiency results are very strong, but the accuracy gains are not always consistent, and the paper itself shows that NOPE* can degrade at high coarsening ratios due to approximation bias.  For example, the variance of NOPE* is larger than NOPE in Figure 2 and Figure 7.

---

> ### Author Rebuttal · Authors · 2026-03-31
>
> W1/Q1: Thank you for your suggestion. Actually, in prior work, such as 'On the Sentence Embeddings from Pre-trained Language Models', has shown that BERT-generated embeddings exhibit a certain degree of anisotropy, which provides empirical support for our assumption and the design of NOPE\*.
> The strong fit of the I curve under the combined NOPE and NOPE\* settings in Figures 2 and 7 particularly in the early and middle stages of the coarsening process, provides compelling empirical support for our hypothesis. This phenomenon of high concordance further corroborates, albeit indirectly, the various assumptions we made in our model.
>
> W2: Thank you for your suggestion. Our use of near-linear refers to the practical regime where the dynamic degree $\Delta$ remains relatively small during the coarsening process ($\Delta$ << n). Under this condition, the overall complexity can be reasonably approximated as linear in n. This behavior is most evident in sparse graphs, where degree growth during merging is naturally limited. When the coarsening process does not significantly increase node degrees, the algorithm exhibits scaling close to linear.
> In cases of aggressive coarsening or graphs with dense local structures, /delta may grow substantially, leading to superlinear behavior. Therefore, NOPE is not strictly linear in general, but approaches linear scaling when /delta remains controlled—particularly in sparse settings.
>
> W3: Thank you for your suggestion. Acutally, there is a clear no-free-lunch trade-off: while NOPE\* significantly improves efficiency, it may introduce slight variability in accuracy due to its approximation. That said, the overall performance gap remains modest in practice. NOPE\* retains competitive accuracy while delivering substantial speedups, making it a favorable choice when efficiency is a priority, whereas NOPE offers greater stability when precision is more critical.
>
> Q2: Thank you very much for your suggestion; The core focus of this research is not on developing a fully engineered solution, but rather on technological exploration. Whatever, we will certainly take your suggestion into account as a direction for our future work.
>
> Q3: Thank you very much for your suggestion. We have added the Delaunay_n24 dataset, which comprises 16 million nodes and 50 million edges. We utilized DeepWalk to generate a 64-dimensional vector for each node, while simultaneously reducing the node scale to 0.3, 0.5, 0.7.
> The NOPE\* coarsening results regarding runtime and memory consumption are presented in the following table. Even when faced with a network of such massive scale, the algorithm maintains exceptional efficiency in terms of both runtime and memory footprint—a fact that fully substantiates its outstanding performance. However, other algorithms, such as A-CM/UGC, are no longer capable of processing such a large dataset within 5 hours and 96GB extra memory.
>
>
> | Ratio | 0.3     | 0.5     | 0.7     |
> |------|---------|---------|---------|
> | Runtime    | 2793.35s | 3350.57s | 4156.46s |
> | Memory   | 3352.97MB | 38159.92MB | 4401.52MB |

---

> > ### Author Rebuttal · Reviewer_udUQ · 2026-04-04
> >
> > Thank you for these responses. I will keep my score unchanged.

---

> > > ### Author Response · Authors · 2026-04-04
> > >
> > > Thank you very much for your response. We will continue to refine our work based on your views. Thank you once again for your support.

---

### Decision · Program_Chairs · 2026-04-30

**Decision:**

Accept (regular)

**Comment:**

After considering the reviews, rebuttal, and discussion, I recommend acceptance. The paper makes a practically meaningful contribution to graph coarsening through an efficient neighborhood-aware merge criterion and a fast approximation, and the strongest part of the submission is its substantial improvement in runtime and memory with broadly competitive downstream performance. While some aspects of the framing are stronger than necessary, particularly around novelty and the approximation assumptions, I do not view these concerns as outweighing the core value of the work. Overall, I believe the paper offers a useful and scalable method that is likely to be of interest to the graph learning community and falls on the accept side of the bar.